# Forecasting weekly dengue incidence in Sri Lanka: Modified Autoregressive Integrated Moving Average modeling approach

**Nilantha Karasinghe**[1], **Sarath Peiris**[2], **Ruwan Jayathilaka**[3]*, **Thanuja Dharmasena**[4]

**1** Teaching Hospital, Nagoda, Kalutara, Sri Lanka, **2** Department of Mathematics and Statistics, Faculty of Humanities and Sciences, Sri Lanka Institute of Information Technology, Malabe, Sri Lanka, **3** Department of Information Management, SLIIT Business School, Sri Lanka Institute of Information Technology, Malabe, Sri Lanka, **4** National Coordinator, Global Environment Facility Small Grants Programme, UNDP, Colombo, Sri Lanka

* ruwan.j@sliit.lk

**Data Availability Statement:** All data in this article are from Weekly Epidemiological Report of Sri Lanka at https://www.epid.gov.lk The authors confirm the authors had no special access

## Abstract

Dengue poses a significant and multifaceted public health challenge in Sri Lanka, encompassing both preventive and curative aspects. Accurate dengue incidence forecasting is pivotal for effective surveillance and disease control. To address this, we developed an Autoregressive Integrated Moving Average (ARIMA) model tailored for predicting weekly dengue cases in the Colombo district. The modeling process drew on comprehensive weekly dengue fever data from the Weekly Epidemiological Reports (WER), spanning January 2015 to August 2020. Following rigorous model selection, the ARIMA (2,1,0) model, augmented with an autoregressive component (AR) of order 16, emerged as the best-fitted model. It underwent initial calibration and fine-tuning using data from January 2015 to August 2020, and was validated against independent 2000 data. Selection criteria included parameter significance, the Akaike Information Criterion (AIC), and Schwarz Bayesian Information Criterion (SBIC). Importantly, the residuals of the ARIMA model conformed to the assumptions of randomness, constant variance, and normality affirming its suitability. The forecasts closely matched observed dengue incidence, offering a valuable tool for public health decision-makers. However, an increased percentage error was noted in late 2020, likely attributed to factors including potential underreporting due to COVID-19-related disruptions amid rising dengue cases. This research contributes to the critical task of managing dengue outbreaks and underscores the dynamic challenges posed by external influences on disease surveillance.

## Introduction

Dengue, a viral ailment, holds significant public health importance and is transmitted by vector mosquitoes, specifically *Aedes aegypti* and *Aedes albopictus*. These vectors are arthropods with a life cycle involving aquatic stages, rendering the transmission of the disease particularly susceptible to climatic factors [1] such as rainfall, humidity, temperature, and wind patterns.

privileges others would not have with the data. All relevant data are within the manuscript and its Supporting Information files.

**Funding:** The authors received no specific funding for this work.

**Competing interests:** The authors have declared that no competing interests exist.

Furthermore, the incidence of dengue is contingent upon the necessity for contact between the vector and the human host, making anthropogenic behavior and host exposure influential factors. The interplay of seasonal variations, encompassing both climatic factors and vector dynamics, is likely to exert a substantial impact on the transmission patterns of the disease. Viral diseases exhibit a wide spectrum of manifestations, ranging from asymptomatic cases to severe illness. It is important to note that only individuals displaying clinical symptoms are typically identified and reported as cases. The virus's capacity to provoke noticeable symptoms in hosts hinges on its aggressiveness, a quality referred to as virulence.

The incidence patterns of dengue are intricate and may exhibit seasonality due to the aforementioned considerations. These distinctive attributes of dengue have spurred many researchers to employ statistical methods for characterising the disease's epidemiology and offering robust projections. Nonetheless, a majority of these investigations tend to be context-specific and confined to local contexts, limiting their broader applicability [2]. Despite this, several authors have endeavuored to elucidate the disease's dynamics and make predictive assessments grounded in observable variables. A significant aspect that has often been overlooked is the quantification of the impact of latent factors such as viral strain virulence, host and vector behavioural patterns, and the influence of herd immunity. The scarcity of such analyses can be attributed to the inherent challenges in precisely quantifying these factors unless sophisticated mathematical modeling is employed [3]. Consequently, several researchers have resorted to employing univariate time series analysis, which involves the examination of individual variables recorded sequentially over time, to address these complexities.

Researchers have employed a variety of methodologies in their pursuit of accurate forecasting. Approaches such as Autoregressive Integrated Moving Average (ARIMA) models, Wavelet Time Series analysis, General Additive Mixed (GAM) models, Spatial analysis, Non-linear methods, Multivariate modeling, and Global Circulation models have all found frequent application in modelling and projecting the occurrence of communicable diseases, including dengue [4]. Among these methods, ARIMA modelling, originally developed by Box and Jenkins, stands out as a widely utilised approach for the statistical forecasting of time series data [5]. One empirical study, for instance, employed ARIMA modelling to predict the incidence of dengue in a specific region of Malaysia based on weekly case data [6]. Given that the interactions between vectors and humans play a pivotal role in shaping dengue dynamics, it is reasonable to postulate that ARIMA models can effectively capture the patterns of dengue incidents from time series data. This is plausible because these patterns inherently reflect the intricate interplay of various external factors [7].

The utilisation of ARIMA modelling presents an effective approach to unveil concealed patterns within a dataset, offering forecasts grounded in historical data. This method has gained prominence for predicting future events by utilising data collected within a predefined temporal framework [8]. Beyond the conventional basic ARIMA and seasonal ARIMA models, ensemble ARIMA models have been employed to model dengue incidence by incorporating patterns from neighbouring regions where dengue cases occur [9]. Accumulating evidence from prior research underscores the indispensable role of ARIMA modelling as a pivotal tool in informing decisions related to disease prevention and control [10, 11]. Furthermore, recognising that the suitability of modelling techniques can vary from one location to another, it becomes apparent that adapting ARIMA models to specific geographic contexts is imperative to effectively forecast the occurrence of dengue [12].

Since the 1960s, dengue fever has emerged as a paramount concern for public health authorities in Sri Lanka. It is a critical arboviral disease that exerts a substantial impact on the population in terms of both morbidity and mortality. Consequently, there is a pressing need for accurate forecasting and vigilant monitoring to effectively trace the dissemination of

dengue. The Ministry of Health (MoH) in Sri Lanka routinely manages dengue and other disease outbreaks on an annual basis. Of particular concern is the severe form of dengue, which can result in a case-fatality rate of up to five percent. During the period from January to April 2023, there were 3,477 reported cases. The financial burden of dengue control and hospitalisation in the Colombo district is significant, estimated at a staggering US$ 3.45 million. A substantial portion of these costs, amounting to 79%, is associated with healthcare staff involved in control activities, with hospitalisation costs contributing to 46% of the total expenses. Particularly, the average cost per hospitalisation varies, ranging from US$ 216 to US$ 609 for pediatric cases and from US$ 196 to 866 for adult cases [13].

Therefore, the forecasting of dengue plays a vital role in achieving various objectives, including the reduction of expenses related to disease control and prevention, the provision of optimal care for hospitalised patients, the enhancement of cost-effective planning for resource allocation, and the mitigation of the adverse impacts associated with the disease. Notably, the Colombo district consistently reports the highest proportion of dengue cases among all districts in Sri Lanka across multiple years. Nevertheless, there has been a notable scarcity of research endeavours dedicated to modelling and predicting the incidence of dengue within the Sri Lankan context.

Typically, dengue exhibits a seasonal pattern due to the association between mosquito breeding and the cyclical nature of rainfall. Nevertheless, climate change has the potential to induce phenological shifts in various species. Consequently, the anticipation of weekly dengue incidence assumes significant importance, particularly for the timely issuance of public warnings and notifications to public health authorities [14]. This is especially pertinent in countries with limited resources, such as Sri Lanka, where it is imperative to gain a comprehensive understanding of the dynamics of dengue transmission to devise more efficient control strategies. Therefore, comprehending the disease's propagation patterns within shorter time intervals, notably on a weekly basis, is crucial for the prevention and management of dengue in Sri Lanka.

The primary aim of this research is to identify a suitable ARMA model for the purpose of characterising and predicting the weekly occurrence of dengue cases within the Colombo district of Sri Lanka. This investigation utilises data from the disease surveillance system managed by the Epidemiology Unit (Epid Unit) under the MoH, which serves as the central authority responsible for the prevention and management of infectious diseases of public health significance. The current study endeavours to address a noteworthy gap in the existing body of research. By doing so, it aims to contribute fresh perspectives and valuable insights concerning the prevention and management of dengue fever in the Colombo district of Sri Lanka. In doing so, it is expected to enhance the understanding of the temporal patterns and dynamics of dengue transmission, thereby supporting more effective strategies for its prevention and treatment in this specific geographic region.

## Materials and methods

### Statistical analysis

The methodology employed for constructing a time series model to analyse the observed data series adhered to the univariate ARMA models recommended within the framework devised by Box-Jenkins [15]. In these models, the conventional ARMA (p, q) configuration comprises a combination of q moving average (MA) and p autoregressive (AR) parameters. This ARMA (p, q) model can be represented as follows,

$$Y_t = \mu + \emptyset_1 Y_{t-1} + \emptyset_2 Y_{t-2} + \cdots + \emptyset_p Y_{t-p} + e_t - \theta_1 e_1 - \theta_2 e_2 - \cdots \theta_q e_{t-q}$$

**Table 1. True pattern of ACF and PACF used for model identification.**

| Model | ACF pattern | PACF Pattern |
|---|---|---|
| AR(p) | Exponential decaying, damped sine wave pattern, or both | Significant spikes through the first lag |
| MA(q) | Significant spikes through the first lag | Exponential decaying |
| ARMA(p,q) | Exponential decaying | Exponential decaying |
| ARMA(l,l) | Exponential decaying from lag l | Exponential decaying from lag l |

Where $\emptyset_i$ (i = 1,2,. . .p) and $\theta_j$ (= 1.2. . ..q) are the coefficient AR and MA parts, respectively, and $\{e_t\}$ is the white noise process. The choice of the most appropriate model was determined in accordance with the procedure advocated by the Box-Jenkins methodology [15]. The initial step in this process involves ensuring the stationarity of the time series [16]. If the original series is non-stationary, stationary is generally achieved by taking $d^{th}$ order differencing such that the series $\{y_t - y_{t-d}\}$ is stationary. Once the stationary analysis was performed by the $d^{th}$ order differencing, the model is denoted by ARIMA(p,d,q). The p and q of ARIMA(p,d,q) are determined by comparison of the pattern of the sample autocorrelation function (ACF) and sample partial autocorrelation function (PACF) with the true ACF and PACF. Generally, few parsimonious models are initially considered. The true pattern of ACF and PACF is shown in Table 1.

From the various hypothesised models, the model that exhibits the best fit is determined through the application of diverse criteria. These criteria encompass the model's overall significance, the significance of individual parameters within the model, and the utilisation of several information criteria. Among these information criteria, the Akaike Information Criterion (AIC) and the Schwarz Bayesian Information Criterion (SBIC) are two widely employed indicators. It is anticipated that these criteria will yield their minimum values for the optimal model selection [16].

$$AIC = -2 \log(\text{likelihood}) + 2k, \text{ and}$$

$$SBIC = -\frac{2}{T} \log(\text{likelihood}) + \log(n) * k, \text{ where } k = \# \text{ } of \text{ parameters}$$

Upon determining the most suitable model, a series of diagnostic assessments were conducted in preparation for the forecasting phase. It was expected that an effective model would exhibit residuals devoid of any discernible systematic patterns. Additionally, the invertibility of the underlying ARMA process was examined by scrutinising the roots of the autoregressive (AR) and moving average (MA) components within the unit circle. Subsequently, the forecasting accuracy of the chosen model was evaluated by comparing it to the actual weekly dengue incidence data, employing both individual percentage error measures and the Mean Absolute Percentage Error (MAPE).

## Secondary data

Dengue fever cases that were documented and reported originate from the Weekly Epidemiological Reports (WER), encompassing the period from 1 January 2015 to 30 December 2020. This investigation relies on secondary data sourced from nationally disseminated weekly surveillance reports published by the MoH in Sri Lanka. The dataset employed for this research is made available in Supplementary S1 Appendix. The statistical analysis for this study was conducted using EViews 12 software.

## Results and discussion

### Temporal variability of the original series

The Fig 1 time series plot visually depicts the temporal fluctuations in the weekly dengue case incidence from January 2015 to August 2020.

As depicted in Fig 1, the weekly incidence of dengue cases exhibits a considerable range, with a minimum of 2 cases recorded during the first week of April 2020 and a maximum of 1972 cases in the third week of July 2017. The central tendency of data is reflected in the mean of 295 cases (standard deviation, SD = 299) and a median of 201 cases. Especially, Fig 1 highlights a period of significantly elevated dengue cases, spanning from the second week of June to the first week of September. It is worth noting that since the country's initial dengue outbreak in 1965–1966, the highest incidence was reported in 2017, with a staggering 186,101 cases. Markedly, more than 25% of the country's dengue cases were concentrated within the Colombo district during this period.

These data points clearly exhibit outlier characteristics when assessed using both the interquartile range method (Q1-1.5IQR, Q3+1.5IQR) and a common heuristic criterion (z-score exceeding 3 or falling below -3). Nevertheless, it is imperative to underscore that, during the formulation of our ARIMA model, we refrained from excluding these data points on the grounds that they represent genuine and observed values integral to the current analysis. Furthermore, as portrayed in Fig 1, the time series plot underscores the non-stationary nature of the original observed series. This assertion was corroborated through the utilisation of the Autocorrelation Function (ACF) plot (Fig 2) and the Augmented Dickey-Fuller (ADF) test [17]. The correlogram of the initial series firmly established its non-stationarity, evidenced by the first few autocorrelations significantly deviating from zero, with a gradual and sustained decline across the lags. The results obtained from the ADF test for the initial series were inconclusive (ADF Test statistic = -2.247, p = 0.135), thereby confirming its non-stationary character.

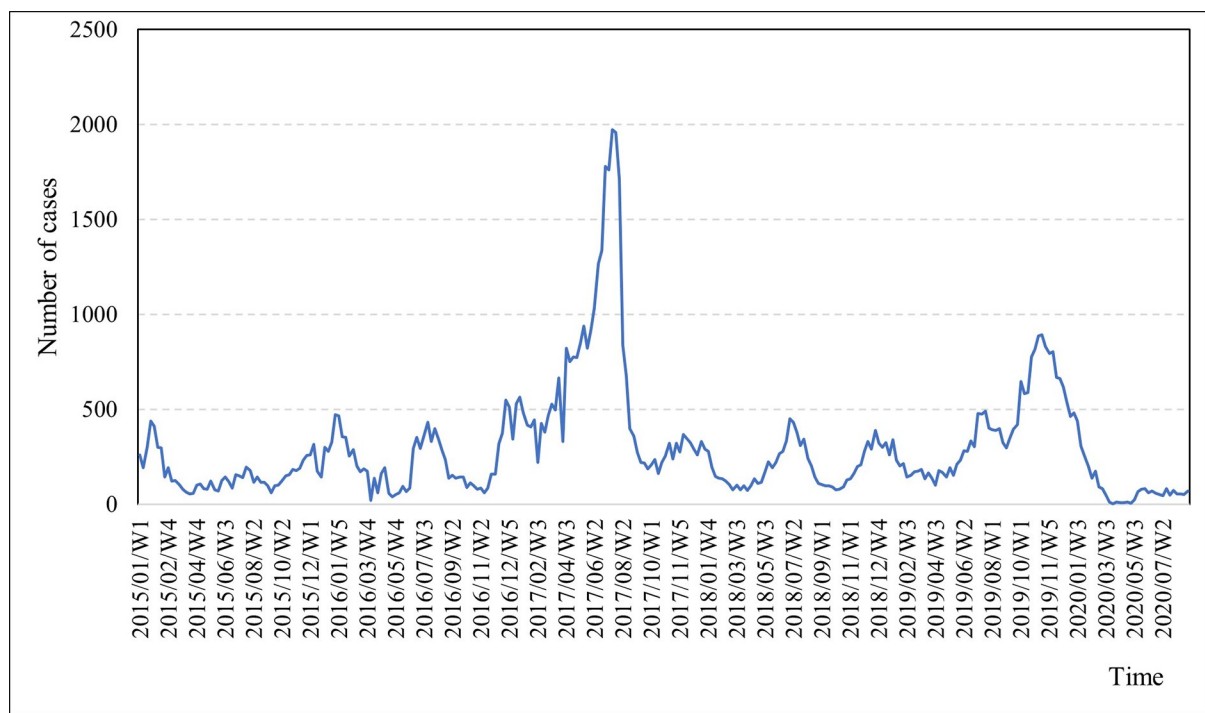

**Fig 1. Time series plot of the weekly incidence of dengue cases.**

| Autocorrelation | Partial Correlation | | AC | PAC | Q-Stat | Prob |
|---|---|---|---|---|---|---|
| | | 1 | 0.947 | 0.947 | 268.18 | 0.000 |
| | | 2 | 0.890 | -0.063 | 506.05 | 0.000 |
| | | 3 | 0.807 | -0.285 | 702.35 | 0.000 |
| | | 4 | 0.719 | -0.093 | 858.59 | 0.000 |
| | | 5 | 0.627 | -0.027 | 977.85 | 0.000 |
| | | 6 | 0.544 | 0.055 | 1067.8 | 0.000 |
| | | 7 | 0.472 | 0.076 | 1135.8 | 0.000 |
| | | 8 | 0.405 | -0.040 | 1186.1 | 0.000 |
| | | 9 | 0.352 | 0.023 | 1224.3 | 0.000 |
| | | 10 | 0.306 | -0.002 | 1253.2 | 0.000 |
| | | 11 | 0.264 | -0.046 | 1274.7 | 0.000 |
| | | 12 | 0.222 | -0.056 | 1290.0 | 0.000 |
| | | 13 | 0.191 | 0.071 | 1301.4 | 0.000 |
| | | 14 | 0.164 | 0.048 | 1309.8 | 0.000 |
| | | 15 | 0.133 | -0.113 | 1315.4 | 0.000 |
| | | 16 | 0.108 | 0.000 | 1319.0 | 0.000 |
| | | 17 | 0.098 | 0.176 | 1322.1 | 0.000 |
| | | 18 | 0.083 | -0.087 | 1324.3 | 0.000 |
| | | 19 | 0.073 | -0.039 | 1326.0 | 0.000 |
| | | 20 | 0.065 | 0.019 | 1327.3 | 0.000 |
| | | 21 | 0.060 | 0.007 | 1328.5 | 0.000 |
| | | 22 | 0.057 | 0.048 | 1329.5 | 0.000 |
| | | 23 | 0.053 | -0.036 | 1330.4 | 0.000 |
| | | 24 | 0.052 | -0.002 | 1331.3 | 0.000 |

**Fig 2. Correlogram of the original series of data.** Note: [AC = Autocorrelation, PAC = partial autocorrelation].

## Differencing the data series

To achieve the stationarity of the data series, first-order differencing was performed. Fig 3 shows the trend in the first differenced series.

Fig 3 exhibited a more pronounced mean reversion pattern, indicating a greater degree of stability over time. Consequently, the first-order differenced data series appeared to exhibit stationarity. This supposition was substantiated by the Augmented Dickey-Fuller (ADF) test conducted on the first differenced series, which revealed statistical significance at a five percent significance level (ADF = -5.277, p = 0.000). Thus, it can be asserted with 95% confidence that the first-order differenced data series is stationary.

## Possible models

To identify the possible parsimonious model for the stationary series, ACF and PACF were obtained, and the correlogram of the first differenced series is shown in Fig 4.

Both ACF and PACF plots, as displayed in Fig 4, revealed a swift decline in values after lag 2, with statistically significant correlations only at that particular lag. This pattern suggests that both the ACF and PACF exhibit an analogous exponential decay characteristic. Consequently, the following three models could be identified, namely ARIMA (2,1,0), ARIMA (0,1,2), and ARIMA (2,1,2), as the most pertinent provisional models to advance with.

## Identification of the best-fitted model

A comparison of those three ARIMA models based on conventional criteria for assessing the goodness is shown in Table 2.

The outcomes presented in Table 2 reveal that all parameters within the ARIMA (2,1,2) model lack statistical significance, rendering it an unsuitable choice for further consideration.

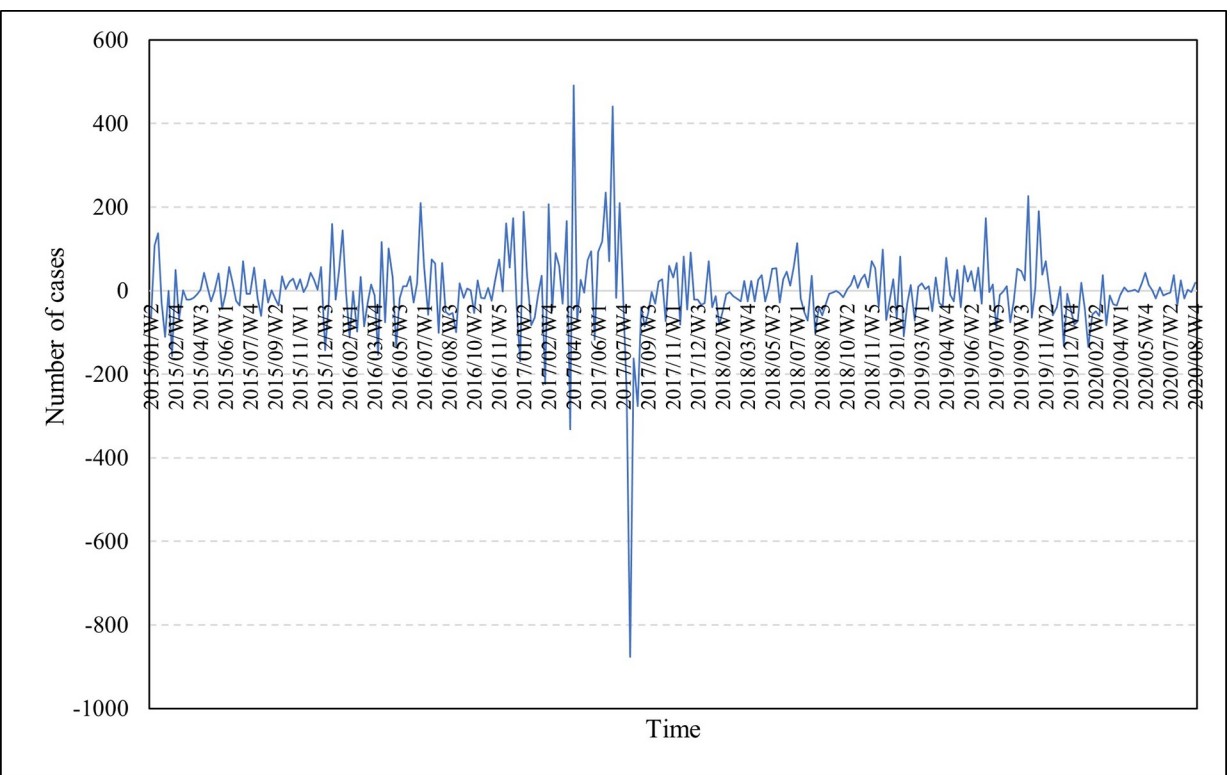

**Fig 3. Time series plot of the first order differencing of the original series.**

Conversely, the results in Table 2 demonstrate that all parameters within the ARIMA (2,1,0) and ARIMA (0,1,2) models are statistically significant, thus establishing both as potential candidates for forecasting. However, given that conspicuous outliers were not excluded from the original series, from a modelling perspective, ARIMA (2,1,0) emerges as the more favourable candidate. This is attributed to its capacity to directly capture the influence of past values, distinguishing it as a more robust choice compared to ARIMA (0,1,2). Furthermore, the $\sigma^2$ volatility in ARIMA (2,1,0) was considerably less than that in ARIMA (0,1,2). Both AIC and SBIC statistics in ARIMA (2,1,0) were less than the corresponding values in ARIMA (0,1,2). Therefore, ARIMA (2,1,0) was chosen as the best-fitted model with comparing forecast values from ARIMA (2,1,0) and ARIMA (0,1,2). The ACF and PACF of the residuals of the ARIMA (2,1,0) are shown in Fig 5.

Fig 5 shows that, up to lag 14, the Q-statistic probabilities for the ACF were not statistically significant. Nevertheless, the correlogram shown in Fig 5 indicates significant correlations after lag 16 in both ACF and PACF. Therefore, a re-estimation was performed by adding AR (16) and MA (16) separately to ARIMA (2,1,0). When assessing these metrics, namely AIC, SBIC, and $\sigma^2$ volatility, it becomes evident that the inclusion of AR (16) terms is more suitable than introducing MA (16) terms. Accordingly, ARIMA (2,1,0) + AR (16) was considered the best-fitted model to represent the weekly incidence of dengue in the tested series. Under these circumstances, it was observed that AR (1) lacked statistical significance. As a result, a revised model was developed, incorporating only AR (2) and AR (16) for the stationary series, as indicated in Table 3.

An examination of the Autocorrelation Function (ACF) plot for the residuals of the best-fitted model affirmed that the errors displayed a random pattern. Furthermore, the plot's

| Autocorrelation | Partial Correlation | | AC | PAC | Q-Stat | Prob |
|---|---|---|---|---|---|---|
| | | 1 | 0.033 | 0.033 | 0.3186 | 0.572 |
| | | 2 | 0.255 | 0.254 | 19.711 | 0.000 |
| | | 3 | 0.053 | 0.042 | 20.565 | 0.000 |
| | | 4 | 0.036 | -0.033 | 20.949 | 0.000 |
| | | 5 | -0.084 | -0.116 | 23.098 | 0.000 |
| | | 6 | -0.110 | -0.121 | 26.782 | 0.000 |
| | | 7 | -0.055 | -0.004 | 27.713 | 0.000 |
| | | 8 | -0.129 | -0.062 | 32.766 | 0.000 |
| | | 9 | -0.065 | -0.034 | 34.078 | 0.000 |
| | | 10 | -0.039 | 0.009 | 34.536 | 0.000 |
| | | 11 | -0.003 | 0.015 | 34.539 | 0.000 |
| | | 12 | -0.100 | -0.111 | 37.610 | 0.000 |
| | | 13 | -0.052 | -0.086 | 38.437 | 0.000 |
| | | 14 | 0.051 | 0.079 | 39.255 | 0.000 |
| | | 15 | -0.064 | -0.039 | 40.546 | 0.000 |
| | | 16 | -0.154 | -0.211 | 48.000 | 0.000 |
| | | 17 | 0.058 | 0.062 | 49.059 | 0.000 |
| | | 18 | -0.050 | 0.015 | 49.862 | 0.000 |
| | | 19 | -0.023 | -0.050 | 50.026 | 0.000 |
| | | 20 | -0.021 | -0.039 | 50.169 | 0.000 |
| | | 21 | -0.027 | -0.078 | 50.406 | 0.000 |
| | | 22 | 0.016 | 0.014 | 50.487 | 0.001 |
| | | 23 | -0.036 | -0.020 | 50.910 | 0.001 |
| | | 24 | 0.077 | 0.019 | 52.815 | 0.001 |

**Fig 4. Correlogram of the first order differenced series.**

representation of residuals, as well as the predicted values, illustrated that the residuals maintain a constant variance. Finally, Fig 6 demonstrates that the distribution of residuals adheres to a normal pattern [18]. An examination of the ACF plot for the residuals of the best-fitted model affirmed that the errors displayed a random pattern. Furthermore, the plot's representation of residuals, as well as the predicted values, illustrated that the residuals maintain a constant variance. Finally, Fig 6 demonstrates that the distribution of residuals adheres to a normal pattern.

The two information criteria used to assess the goodness of fit in the best-fitted model yielded values of 11.91198 and 11.96197, respectively. The mean absolute percentage error (MAPE) measured at 0.3184, suggesting that the fitted model is well-suited for forecasting purposes. The Theil Inequality Coefficient (U) registered at 0.108, with an exceedingly small bias

**Table 2. Comparison of the selected ARIMA models.**

| Indicators | Model | | |
|---|---|---|---|
| | ARIMA (2,1,0) | ARIMA (0,1,2) | ARIMA (2,1,2) |
| Parameter–AR (1) | significant | not applicable | not significant |
| Parameter–AR (2) | significant | not applicable | not significant |
| Parameter–MA (1) | not applicable | significant | not significant |
| Parameter–MA (2) | not applicable | significant | not significant |
| $\sigma^2$_Volatility | 8743.8 | 8770.2 | 8738.7 |
| AIC | 11.935 | 11.938 | 11.940 |
| SBIC | 11.972 | 11.975 | 11.991 |

| Autocorrelation | Partial Correlation | | AC | PAC | Q-Stat | Prob |
|---|---|---|---|---|---|---|
| | | 1 | 0.014 | 0.014 | 0.0566 | |
| | | 2 | 0.008 | 0.008 | 0.0776 | 0.781 |
| | | 3 | 0.075 | 0.075 | 1.7645 | 0.414 |
| | | 4 | 0.001 | -0.001 | 1.7650 | 0.623 |
| | | 5 | -0.095 | -0.097 | 4.5126 | 0.341 |
| | | 6 | -0.100 | -0.105 | 7.5691 | 0.182 |
| | | 7 | -0.022 | -0.019 | 7.7194 | 0.259 |
| | | 8 | -0.106 | -0.091 | 11.147 | 0.132 |
| | | 9 | -0.059 | -0.043 | 12.199 | 0.143 |
| | | 10 | 0.018 | 0.015 | 12.301 | 0.197 |
| | | 11 | 0.028 | 0.026 | 12.545 | 0.250 |
| | | 12 | -0.117 | -0.128 | 16.773 | 0.115 |
| | | 13 | -0.040 | -0.069 | 17.282 | 0.139 |
| | | 14 | 0.127 | 0.101 | 22.335 | 0.050 |
| | | 15 | -0.075 | -0.072 | 24.089 | 0.045 |
| | | 16 | -0.176 | -0.190 | 33.770 | 0.004 |
| | | 17 | 0.090 | 0.056 | 36.305 | 0.003 |
| | | 18 | -0.010 | -0.024 | 36.335 | 0.004 |
| | | 19 | -0.034 | -0.015 | 36.706 | 0.006 |
| | | 20 | -0.015 | -0.048 | 36.775 | 0.008 |
| | | 21 | -0.015 | -0.077 | 36.848 | 0.012 |
| | | 22 | 0.003 | -0.003 | 36.851 | 0.018 |
| | | 23 | -0.060 | -0.064 | 37.991 | 0.018 |
| | | 24 | 0.071 | 0.013 | 39.642 | 0.017 |

**Fig 5. Correlogram of the residuals of the ARIMA model (2,1,0).**

proportion of 0.000008. Given the proximity of U to 0, this further underscores the model's strong predictive capabilities [19]. Examination of the AR roots indicated that they are located within the unit circle, confirming the stability of the final model. Taken together, these results provide compelling evidence of the robustness and stability of the ARIMA (2,1,0) + AR (16) model.

A comparison between the observed and predicted values for both the training and validation datasets (out of the sample) is illustrated in Figs 7 and 8, respectively.

**Table 3. Details of the parameters of ARIMA (2,1,0) + AR (16).**

| Variable | Coefficient | Std. Error | t-Statistic | Prob |
|---|---|---|---|---|
| C | -0.529103 | 7.472788 | -0.070804 | 0.9436 |
| AR (2) | 0.262049 | 0.054137 | 4.840446 | 0.0000 |
| AR (16) | -0.16328 | 0.027749 | -5.884273 | 0.0000 |
| SIGMASQ | 8475.277 | 308.0737 | 27.51055 | 0.0000 |
| S.E. of regression | 92.69183 | Akaike info criterion | | 11.91198 |
| Sum squared residual | 2500207 | Schwarz Bayesian Info criterion | | 11.96197 |
| Log-likelihood | -1753.017 | Hannan-Quinn criterion | | 11.93200 |
| F-statistic | 10.0565 | Durbin-Watson statistic | | 1.960523 |
| Prob(F-statistic) | 0.000003 | | | |

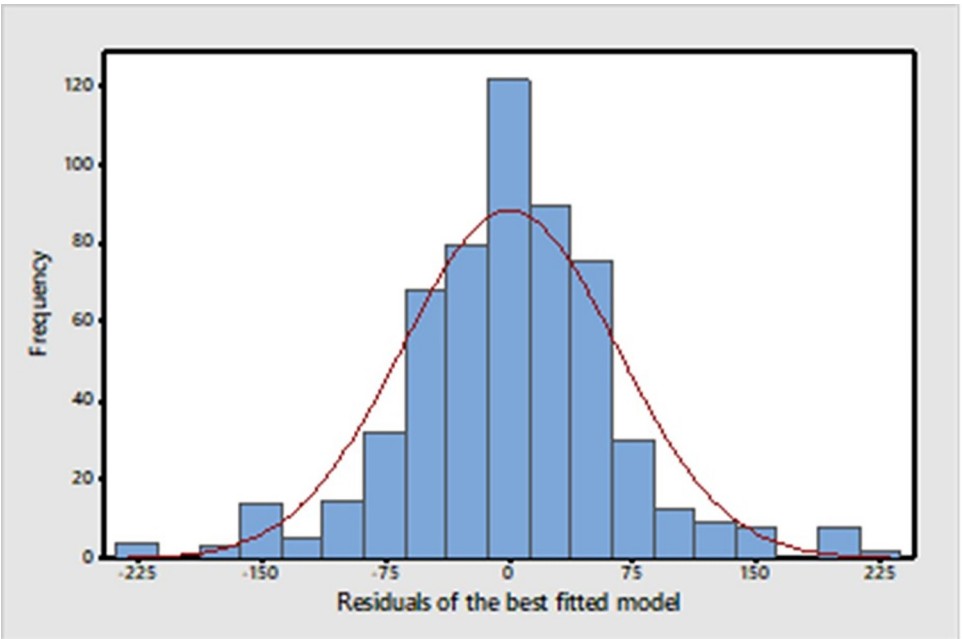

**Fig 6. Distribution of the residuals of the best fitted models.**

Fig 7 demonstrates a noteworthy level of consistency between the observed and predicted values. This observation is substantiated by a robust and statistically significant correlation (r = .954, p = 0.000) between the actual and forecasted values. This correlation signifies that the

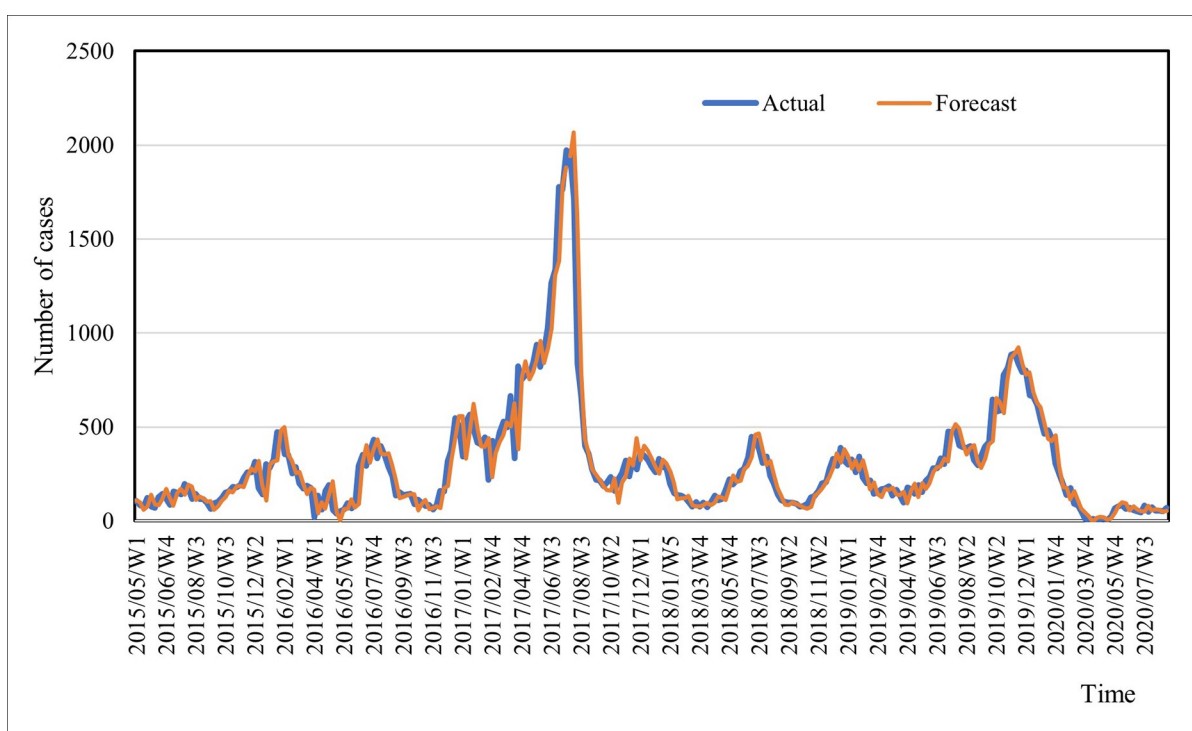

**Fig 7. Forecasted and actual values of the incidence of dengue for the training data set.**

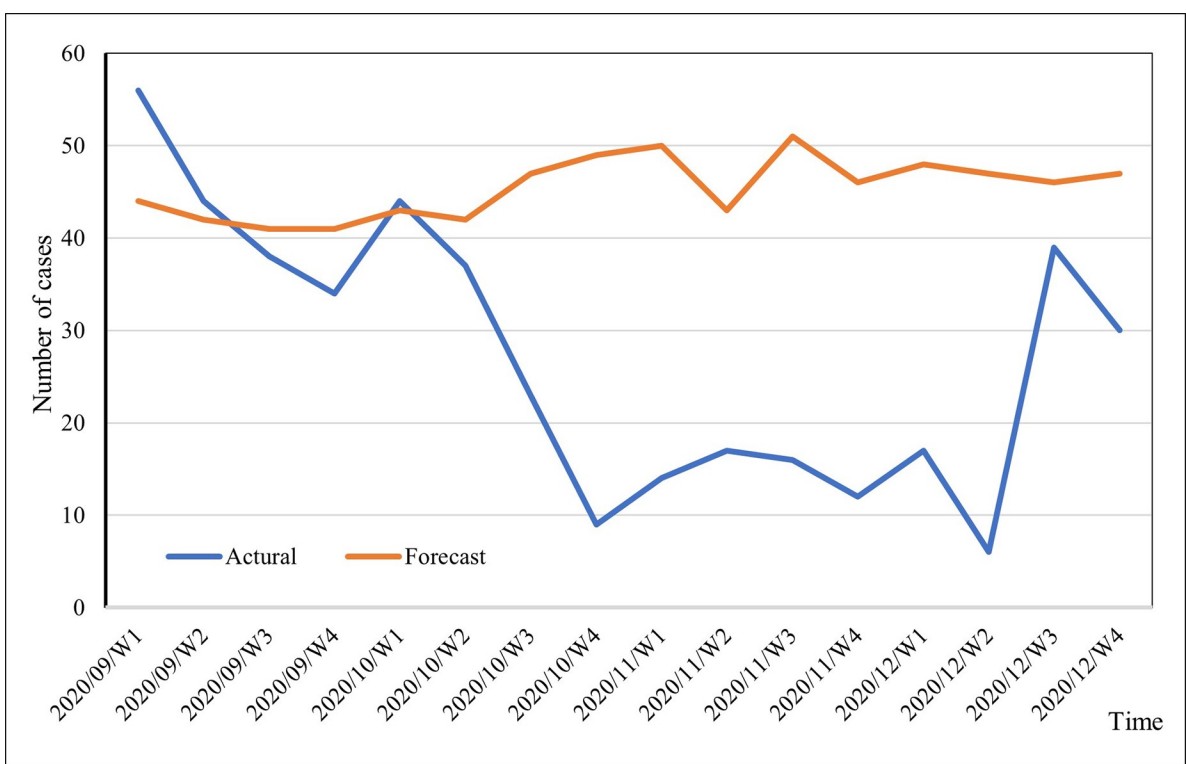

**Fig 8. Forecasted and actual values of the incidence of dengue for the validation set (September—December 2020).**

model's predictive pattern effectively captures the frequency of dengue cases. In Fig 8, the graph displays the forecasted and actual incidence of dengue from September to December 2020, thereby representing a short-term prediction.

The MAPE for the validation set is 1.554, which is markedly higher than the MAPE of 0.3184 for the training set. Additionally, the correlation between the actual and predicted values in the validation set is -0.662 (p = 0.005). These findings suggest that the new model exhibits superior performance when applied to the training dataset in comparison to the validation set. While there are instances where the predictions seem to overestimate the actual values in the validation set, the overall pattern of the model remains suitable for forecasting dengue incidence. However, it is significant that the model tends to overestimate cases post-October 2020, with percentage errors exceeding 100%. This particular period, covering early September 2020 to mid-December 2020, coincided with the effects of the COVID-19 pandemic in Sri Lanka. During this time, factors such as reduced social mobility, disruptions in human activities (especially in dengue-prone environments), and the allocation of limited resources for pandemic management likely contributed to an underreporting of dengue cases. Public health officials may have been compelled to prioritise pandemic management over dengue control due to factors such as cost constraints, vaccine distribution, and strained healthcare resources.

Vector-borne diseases result from the transmission of parasites, bacteria, or viruses through intermediary hosts, such as mosquitoes, ticks, and fleas. In addition to dengue, common vector-borne diseases prevalent in Sri Lanka include chikungunya, yellow fever, Zika virus, and Japanese encephalitis. According to the World Health Organisation (WHO) [20], vector-borne diseases account for approximately 17% of all infectious diseases and are responsible for over 700,000 deaths annually. These diseases are most prevalent in tropical and subtropical

regions, like Sri Lanka, where recent years have witnessed outbreaks of dengue, malaria, yellow fever, and Zika virus in numerous countries.

ARIMA models with seasonal multiplication (SARIMA) have been applied for malaria forecasting across different districts in Sri Lanka. In the context of short-term malaria prediction models, an empirical investigation has highlighted significant variations in the forecasting errors of SARIMA models among these districts [21]. The authors emphasised that while the inclusion of rainfall as a covariate could enhance the predictive accuracy of SARIMA models in certain districts, it could conversely lead to deteriorated predictions in other Sri Lankan districts. It is worth noting that in this study as well, individual prediction errors have not been calculated.

There are four dengue virus serotypes designated as DENV 1–4. Each of these serotypes has dominated dengue outbreaks in specific years [22]. Given that the severity of an epidemic often correlates with the prevailing serotype, mathematical modelling can serve as a valuable instrument for formulating hypotheses regarding the emergence of new strains.

Informed decision-making is consistently bolstered by precise forecasting. Diseases with a substantial public health impact, such as dengue, can benefit from the application of mathematical and statistical models, particularly ARIMA models. Furthermore, these models are poised to play a pivotal role in optimising the allocation of limited resources and maintaining the integrity of public health services in Sri Lanka, particularly during critical periods. Anticipating the number of cases in advance holds the potential to reduce the substantial outflows of foreign currency associated with the importation of medications.

## Contribution of this study

In Sri Lanka, the risk of dengue fever is pervasive and extends throughout the year. Nevertheless, the transmission rates of the disease typically reach their peak during the two monsoon seasons in May/June and October/November. The MoH recognises dengue fever and other mosquito-borne illnesses as growing public health concerns within the country. Sri Lankan health authorities have recently concentrated their efforts on the development of an "early warning system" aimed at reducing dengue incidence rates and efficiently allocating limited public health resources to successful intervention programs. Over the past decade, several local researchers [23–25] have undertaken investigations to forecast dengue incidents at various temporal scales. Most of these efforts have revolved around regression models and univariate ARMA models. However, dengue is the result of intricate interactions among vectors, pathogens, and human populations. Consequently, a multitude of factors, encompassing ecological, environmental, epidemiological, and social elements, influence the distribution patterns of dengue outbreaks. These distribution patterns and epidemic cycles can undergo transformations when novel vector control strategies and policies are implemented. Consequently, the process of selecting suitable variables for regression models becomes inherently challenging.

Furthermore, it is essential to acknowledge that due to the dynamic nature of the influencing variables described earlier, traditional regression models may face limitations in generating precise forecasts. When juxtaposed with regression models, ARIMA models emerge as a more robust and dependable approach. This is because ARIMA models inherently account for the effects of these external factors by indirectly incorporating them into the time series modelling process.

The validation of the model, both against the training dataset and an independent dataset, represents a distinctive feature of the ARMA model developed in this current study. Most research that endeavours in this domain have typically omitted the testing of models against separate datasets. While the selection of the best-fitting model has traditionally relied on

statistical indicators like AIC, SBIC, and MAPE, comparisons between forecasted values and the corresponding observed data have often been neglected. This study addresses these gaps by establishing the ARIMA model as a dependable forecasting tool, particularly in the context of the rapid proliferation of dengue cases in the Colombo district, Sri Lanka. In this study, an additional aspect was considered, namely, the percentage errors associated with each observed data value. The authors of this study deemed this aspect crucial from both a modelling and management perspective. Furthermore, the model devised in this research deviates from classical ARMA models. To enhance the fit of the ARIMA model, an unconventional approach was taken: the introduction of an exclusive secondary parameter, AR (16). This addition was made distinctly for AR (16) and MA (16) separately. Based on the available literature, no similar methods have been identified. Consequently, it is reasonable to assert that this study represents an inaugural exploration, verifying the ARIMA model as a reliable instrument for dengue forecasting.

## Limitations of the study

The study has several limitations. First, the quality and completeness of the data are crucial, as inaccuracies or missing data in the historical dengue records can bias the model and reduce its effectiveness. Second, the introduction of the AR (16) term in the ARIMA model adds complexity, potentially affecting its interpretability and real-world usability. The assumption of stationarity post-differencing may not always be straightforward, impacting model performance. The study exclusively relies on time series data and overlooks external factors like climate and public health interventions, limiting model accuracy. The relatively short validation period could be extended for a more robust model. Generalisability is limited to the Colombo district, and the model's applicability elsewhere is uncertain. The study doesn't explicitly address outlier handling, a critical aspect in time series forecasting, and provides minimal discussion on parameter interpretation, essential for public health decision-makers. The cutoff of data in December 2020 may not consider recent dengue dynamics or public health changes. The selection criteria of the model rely on statistical measures, overlooking clinical relevance and practical utility. Moreover, within the scope of this research, projections were executed at a consistent geographical level, specifically confined to the Colombo district. This approach contrasts with the work undertaken by [26], where time series models were formulated for forecasting dengue incidents, transcending country-level, health district, and state-level considerations. It is noteworthy that, unlike COVID, dengue is not equally prevalent across all districts in Sri Lanka; its primary occurrence is concentrated in the Colombo and Gampaha districts. To facilitate the development of a unified model applicable to both districts a dummy variable can be considered along with weekly dengue incidents for Gampaha district. Finally, the study does not fully account for inherent uncertainty and variability in disease transmission. Discussing prediction intervals and uncertainty measures would provide a more comprehensive view of the performance of the model. Addressing these limitations will enhance the accuracy and suitability of the model for dengue forecasting in Sri Lanka.

## Policy implementation

Dengue stands as the most critical vector-borne disease in Sri Lanka, commanding heightened attention from health authorities who prioritize its control. The effective management of an infectious disease like dengue hinges on the establishment of a robust disease surveillance system. It is imperative to recognise that this surveillance system is susceptible to compromise when concurrent health events of considerable magnitude, characterised by substantial healthcare expenses and vaccine-related costs, such as the COVID-19 pandemic, enter the healthcare

landscape. From an economic and policy perspective, this can lead to a weakened disease surveillance system, potentially resulting in increased healthcare expenditures and higher fatality rates. Consequently, health authorities must take proactive measures to mitigate the impact of these significant health events on disease surveillance, ensuring its resilience. Given the typical one-year lag period associated with health data, forecasting emerges as an ideal technique for the early detection of deviations in health patterns. Such early identification equips health authorities with the capability to allocate resources in advance, guided by data-driven optimisation strategies. This proactive approach facilitates the effective management of health resources and contributes to a more efficient and responsive healthcare delivery system.

## Conclusion

Dengue cases in Sri Lanka have consistently exhibited the highest incidence within the Colombo district across all recorded years. Given this persistent pattern, acquiring advanced insights into the short-term distribution of dengue cases becomes imperative for the MoH to institute timely preventive measures and interventions. In this context, the application of the Box–Jenkins methodology, specifically the ARIMA (2,1,0) model supplemented with an additional AR (16) term, emerges as a robust statistical model. This model has demonstrated its capacity for effectively forecasting weekly dengue cases, making it a viable proposition for short-term forecasting of weekly dengue incidents within the confines of the Colombo district. The utilisation of this model holds the potential to enhance the preparedness and response strategies of MoH, ultimately contributing to the proactive management of dengue outbreaks in the region.

The identification of the autoregressive order stands as a pivotal step in ARIMA modelling, one that significantly influences the effectiveness of the model. The conventional practice suggests the use of low values for both p and q, typically in the range of 0 to 2, based on the observed theoretical pattern of the ACF and PACF of the stationary series. However, when it comes to determining the theoretical ACF for a higher order of autoregressive terms, this becomes a challenging task, especially when only a limited number of higher-order autocorrelations are discernible. In the context of the present study, the authors have introduced a novel approach for the determination of the autoregressive order. This approach centres around the identification of a specific higher-order autocorrelation that exhibits significance. Consequently, the ARIMA (2,1,0) model, augmented by the inclusion of an AR (16) term, has emerged as a robust statistical model for the purpose of forecasting dengue cases. Notably, the incorporation of the AR (16) term represents a distinctive aspect of this approach, setting it apart as an innovative method. To validate the performance and reliability of the model, it was rigorously tested against an independent dataset, confirming its goodness and suitability for accurate dengue case forecasting. This methodological innovation serves as a valuable addition to the field of time series modelling for infectious disease prediction.

Furthermore, it is noteworthy that the model's errors exhibit the characteristics of white noise, emphasising the suitability and reliability of ARIMA (2,1,0) with AR (2) and the supplementary inclusion of AR (16) for forecasting the incidence of dengue in the Colombo district. This model is instrumental in optimising responses to outbreak management, particularly in predicting potential dengue outbreaks within the district, addressing a quintessential concern in ARIMA modelling. However, it is imperative to acknowledge that certain challenges persist in this modelling approach. While the incorporation of AR (16) has demonstrated superiority over alternative models, it is associated with specific limitations, such as underestimating the training set values while overestimating the validation set values. As a result, future research endeavours should concentrate on refining and enhancing methods to mitigate these errors.

These models warrant continuous refinement to rectify their shortcomings. Moreover, the refinement and development of such models hold particular significance during the high-risk periods of May/June and October/December when dengue occurrences reach their peak. The ability to generalise these findings to the broader Sri Lankan context would be immensely beneficial in the realm of comprehensive disease surveillance and health policy planning. Consequently, improving these models and addressing their limitations is an essential undertaking to bolster the effectiveness of dengue outbreak prediction and management in Sri Lanka.

## Supporting information

**S1 Appendix. Data file.**
(XLSX)

## Author Contributions

**Conceptualization:** Nilantha Karasinghe, Sarath Peiris, Ruwan Jayathilaka.

**Data curation:** Nilantha Karasinghe.

**Formal analysis:** Nilantha Karasinghe, Sarath Peiris, Ruwan Jayathilaka.

**Methodology:** Nilantha Karasinghe, Sarath Peiris, Ruwan Jayathilaka.

**Software:** Nilantha Karasinghe.

**Supervision:** Sarath Peiris, Ruwan Jayathilaka.

**Validation:** Nilantha Karasinghe, Sarath Peiris, Ruwan Jayathilaka.

**Visualization:** Nilantha Karasinghe.

**Writing – original draft:** Nilantha Karasinghe, Sarath Peiris, Ruwan Jayathilaka, Thanuja Dharmasena.

**Writing – review & editing:** Nilantha Karasinghe, Sarath Peiris, Ruwan Jayathilaka, Thanuja Dharmasena.

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
