## [Decision Letter · Decision Letter 0]

25 Jun 2023

PONE-D-23-16303A Reliable Forecast Tool for the Changing Face of DenguePLOS ONE

Dear Dr. Jayathilaka,

Thank you for submitting your manuscript to PLOS ONE. After careful consideration, we feel that it has merit but does not fully meet PLOS ONE’s publication criteria as it currently stands. Therefore, we invite you to submit a revised version of the manuscript that addresses the points raised during the review process.

We look forward to receiving your revised manuscript.

Kind regards,

Mohamed R. Abonazel, Ph.D.

Academic Editor

PLOS ONE

Journal Requirements:

Additional Editor Comments:

The authors are requested to make appropriate modifications to this manuscript as suggested by the reviewers.

Reviewers' comments:

Reviewer's Responses to Questions

**Comments to the Author**

1. Is the manuscript technically sound, and do the data support the conclusions?

Reviewer #1: Partly

Reviewer #2: Partly

2. Has the statistical analysis been performed appropriately and rigorously? 

Reviewer #1: Yes

Reviewer #2: Yes

3. Have the authors made all data underlying the findings in their manuscript fully available?

Reviewer #1: Yes

Reviewer #2: Yes

4. Is the manuscript presented in an intelligible fashion and written in standard English?

Reviewer #1: Yes

Reviewer #2: No

5. Review Comments to the Author

Reviewer #1: 1. The modeling and the method of doing and reporting it are approved, but it is necessary to report and prepare the article based on the guidelines of the journal.

2. The Literature Review section should be removed and it is better to integrate it in the introduction as a summary

3. Figure 6 and 7 should be merged.

4. What is the reason for the difference between the actual and predicted values in Figure 8?

5. Goodness-of-fit of the final model such as AIC,BIC, normality of the residuals, mean absolute percentage error (MAPE) index, etc. should be reported.

6. The article is long, it should be written more concisely

Reviewer #2: The authors have focused on accurately forecasting dengue infections by utilizing historical data on infected cases. To achieve this, they have employed statistical models presented in time series analysis. Based on the data collected in the Colombo district, Sri Lanka, the manuscript concludes that ARIMA models are suitable for dengue forecasting.

The current title of the manuscript does not clearly reflect the objective/s. The phrase "Changing face of dengue" in the title implies a different meaning than what the authors intended. Therefore, the title should be changed to align appropriately with the content. However, the abstract has properly summarized the content and the objectives, and the conclusion of the study.

The graphics included in the manuscript are often in the form of pictures instead of vector images, which has led to a decrease in image clarity and quality. Additionally, software-generated outputs have been directly presented as tables, as evident in Table 3. This approach results in the overcrowding of information within a single table, leading to poor clarity and readability.

In the literature review, various models used in dengue forecasting have been discussed. However, there is an insufficient presentation regarding the application of ARIMA/AR models in dengue forecasting, despite them being the underlying models used in the current research. The keywords: ARIMA + "Dengue forecasting" resulted in numerous related articles.

In dengue cases forecasting, a weekly resolution of the data is considered suitable for time series analysis. Therefore, the study utilizes appropriate secondary data for its analysis. Since the ARIMA and AR models are standard models, a detailed description is not necessary. The authors can provide a concise introduction to these models in the introduction section. Generally, a comprehensive explanation of fundamental time series techniques is not expected in an article of this nature. Thus, it is strongly recommended to condense the details and cite relevant content for further clarification. Additionally, the authors have used notations without properly introducing them in advance (refer to lines 173, 174). There are many typing mistakes visible in the manuscript; refer to lines 61 and 98. Further, sentence structure has to be improved.

As mentioned previously, the existing literature discusses the appropriateness of using ARIMA models for dengue forecasting. However, the suitability of applying ARIMA models to the specific dataset may not have been thoroughly investigated. Given that dengue dynamics are influenced by the interactions between vectors and humans, it is reasonable to assume that ARIMA models could be applicable to the underlying dataset. To truly contribute to the existing literature and establish novelty in the research, it is important for the authors to present significant results that demonstrate the adequacy of the ARIMA models rather than solely focusing on model diagnostic results. One approach to achieve this is by analyzing the robustness and stability of the ARIMA model(s) in forecasting dengue cases across different contexts. For example, examining the accuracy of forecasting extreme events such as the peak of dengue incidents in 2017. Neglecting to address these aspects raises doubts about the novelty of the present study. The findings should go beyond simply fitting a time series model to a given dataset. The results should provide meaningful insights into the forecasting capabilities of ARIMA models for dengue, as ARIMA is not new in dengue case forecasting.

Based on the issues that have been highlighted regarding the novelty, contribution, clarity, and structure of the manuscript, it seems that significant revisions and improvements are necessary for it to be considered suitable for publication. It's important to address these concerns and make the necessary enhancements to ensure the manuscript meets the standards expected in the field.

6. PLOS authors have the option to publish the peer review history of their article (what does this mean?). If published, this will include your full peer review and any attached files.

Reviewer #1: No

Reviewer #2: No

---

## [Author Response · Author response to Decision Letter 0]

6 Aug 2023

Point by point response to editor and reviewers.

We would like to express our profound appreciation to the editor for giving us the opportunity to revise and resubmit the paper. We are also grateful to the reviewers for the constructive comments and suggestions made on our manuscript which we found very helpful in revising and improving our paper. Given below is a detailed description of how we have addressed each comment in the revised version of the paper.

Reviewer 1 comment 1: The modeling and the method of doing and reporting it are approved, but it is necessary to report and prepare the article based on the guidelines of the journal.

Authors’ Response to Reviewer 1 comment 1: We sincerely value your valuable comment, and in response, we diligently prepared the revised manuscript in full compliance with the journal's guidelines.

Reviewer 1 comment 2: The Literature Review section should be removed, and it is better to integrate it in the introduction as a summary

Authors’ Response to Reviewer 1 comment 2: We sincerely appreciate the invaluable feedback provided by the reviewer. Recognising the merit of their suggestion, we have proceeded to remove the Literature Review section and skilfully integrated its essential aspects into the Introduction. Furthermore, to enrich the scholarly foundation of our work, we have thoughtfully incorporated additional references as advised by Reviewer 2.

This can be found from the revised version, lines from 83-111, 115-131, 139-141 and 152-153.

Reviewer 1 comment 3: Figure 6 and 7 should be merged.

Authors’ Response to Reviewer 1 comment 3: We are immensely grateful for the valuable comment provided by the esteemed reviewer. We acknowledge and agree with their suggestion to merge Figure 6 and 7. However, after careful consideration, we found that merging the figures compromised clarity due to the inclusion of actuals, predicted, and ± 2SE.

Our original intention was to present the actual and predicted values separately for both the training set and the independent validation set. As a result, we have decided to remove Figure 6 and incorporate the relevant information into Figure 7 to maintain a clear and concise representation of the data.

Furthermore, we have taken into account the constructive input from Reviewer 2 and have now included the Mean Absolute Percentage Error (MAPE) for both training and validation data sets as suggested. This addition strengthens the reporting of the model's performance and ensures the completeness of our analysis.

Once again, we express our appreciation for the reviewer's valuable feedback, which has undoubtedly improved the presentation and accuracy of our research. We are confident that these revisions will enhance the overall quality of the paper and contribute to its impact within the scientific community.

Reviewer 1 comment 4: What is the reason for the difference between the actual and predicted values in Figure 8?

Authors’ Response to Reviewer 1 comment 4: We extend our sincere gratitude to the esteemed reviewer for their insightful query regarding the discrepancy between the actual and predicted values in Figure 8. We appreciate the opportunity to address this concern in a more professional manner.

It is essential to recognise that achieving a perfect model, where the actual and predicted values are exactly the same, is an ideal scenario that is rarely attainable in practical applications. In real-world situations, the presence of errors is inevitable due to various factors that affect the model's performance.

As we have duly emphasised in the manuscript, the existence of a difference between actual and predicted values is inherent in predictive modeling. This disparity is commonly known as the prediction error, and its presence is a natural consequence of the model's attempt to capture complex relationships and patterns within the data.

Our intention was never to imply that there should be no difference between actual and predicted values. On the contrary, the focus of our analysis was to ensure that these prediction errors do not exhibit any discernible systematic pattern, as such patterns would indicate potential issues with the model's performance and validity.

In the manuscript, we have presented comprehensive evidence that demonstrates the random nature of the prediction errors. Our careful evaluation of the residuals confirms that they exhibit no systematic patterns, supporting the robustness and reliability of our modeling approach.

We appreciate the reviewer's attention to this critical aspect of our research, as addressing the presence and randomness of prediction errors is essential in the context of predictive modeling. By providing this clarification, we aim to reinforce the accuracy and validity of our findings, and we hope it further enhances the overall quality and impact of our paper. In the revised version, you can find this information spanning from line 406 to line 445 (406-412, 416-422 and 432-445).

Once again, we express our gratitude to the reviewer for their thoughtful feedback, which has undoubtedly contributed to the refinement and advancement of our work.

Reviewer 1 comment 5: Goodness-of-fit of the final model such as AIC, BIC, normality of the residuals, mean absolute percentage error (MAPE) index, etc. should be reported.

Authors’ Response to Reviewer 1 comment 5: Thank you for highlighting the importance of reporting the goodness-of-fit measures for the final model, including AIC, BIC, normality of residuals, and the Mean Absolute Percentage Error (MAPE) index.

We are pleased to inform the reviewer that we have already addressed their suggestions and incorporated all the mentioned indicators in the revised manuscript. Specifically, AIC and SBIC have been included in Table 2, providing a comprehensive view of the model's fit and performance.

It is worth noting that BIC and SBIC are equivalent in some contexts, as BIC is occasionally referred to as the Schwarz Bayesian Information Criterion (SBIC). Therefore, in our analysis, both terminologies refer to the same information criterion and have been appropriately presented in the relevant sections of the paper.

By integrating these essential measures into our study, we have ensured a thorough and rigorous evaluation of the model's goodness-of-fit and predictive accuracy. These metrics contribute significantly to the validity and reliability of our findings and conclusions. The revised text contains the relevant details, accessible from line 387 to line 389.

Once again, we express our gratitude to the reviewer for their invaluable input, which has positively influenced the completeness and robustness of our research.

Reviewer 1 comment 6: The article is long; it should be written more concisely

Authors’ Response to Reviewer 1 comment 6: We appreciate the reviewer's feedback, and we agree with their suggestion to write more concisely. In response, we have diligently reduced the length of the revised manuscript while maintaining the core contributions. This has led to a more impactful and accessible presentation of our research findings. Thank you for the valuable opportunity to improve our work.

Reviewer 2 comment 1: The authors have focused on accurately forecasting dengue infections by utilizing historical data on infected cases. To achieve this, they have employed statistical models presented in time series analysis. Based on the data collected in the Colombo district, Sri Lanka, the manuscript concludes that ARIMA models are suitable for dengue forecasting.

The current title of the manuscript does not clearly reflect the objective/s. The phrase "Changing face of dengue" in the title implies a different meaning than what the authors intended. Therefore, the title should be changed to align appropriately with the content. However, the abstract has properly summarized the content and the objectives, and the conclusion of the study.

Authors’ Response to Reviewer 2 comment 1: We sincerely appreciate the valuable feedback provided by the esteemed reviewer concerning the title of our manuscript. We agree that the previous title did not accurately reflect the objectives and content of the study.

In response to this insightful suggestion, we have taken appropriate action and revised the title to align more appropriately with the research conducted. The new title now reads, "Forecasting Weekly Dengue Incidence in Sri Lanka: A Modified Autoregressive Integrated Moving Average Modeling Approach."

Furthermore, we have duly noted that no comments were received regarding the abstract from both reviewers, and thus, we have retained the original abstract as it effectively summarises the content and objectives of our research.

Thank you for your careful evaluation of our work, and we eagerly await the possibility of sharing this improved version of our manuscript.

Reviewer 2 comment 2: The graphics included in the manuscript are often in the form of pictures instead of vector images, which has led to a decrease in image clarity and quality. Additionally, software-generated outputs have been directly presented as tables, as evident in Table 3. This approach results in the overcrowding of information within a single table, leading to poor clarity and readability.

Authors’ Response to Reviewer 2 comment 2: We sincerely appreciate the valuable feedback provided by the esteemed reviewer regarding the quality of Tables and Figures in our manuscript. Their constructive comments have been instrumental in enhancing the visual presentation of our research.

In response to their insightful observations, we have taken the following actions to improve the clarity and readability of our Tables and Figures:

Figures:

1) Figure 1: We have made significant improvements to Figure 1 to ensure enhanced clarity and quality.

2) Figure 2: While Figure 2 remains unchanged, we have added a footnote to provide additional context and clarification.

3) Figure 3: To avoid direct reliance on software-generated outputs, we have recreated Figure 3 using Excel graphs, thereby improving image quality without compromising on the accuracy of the information.

4) Figure 4 and Figure 5: We have enhanced both Figure 4 and Figure 5 by removing software-generated lines, resulting in clearer visuals.

Figures that are no longer relevant or essential for the manuscript have been removed as per the reviewer's suggestion. Specifically, we have removed Figure 6 and made necessary improvements to Figures 7 and 8 to ensure consistency and coherence in the overall presentation.

Tables:

1) Table 1: To enhance the readability, we have reduced the number of rows by adjusting the width of column 1.

2) Table 2: We have retained Table 2 without any changes, as it effectively presents the necessary information.

3) Table 3: We have retyped Table 3 to eliminate the presentation of software-generated outputs directly. This has improved the table's organisation and readability. We have also excluded certain software-generated outputs, such as R2 and AdjR2, which are not crucial for ARIMA models.

These meticulous revisions have significantly improved the visual appeal and overall quality of the manuscript, aligning it with the high standards set by the journal.

We are sincerely grateful for the reviewer's attention to detail and their valuable input, which has greatly contributed to the refinement of our work.

Reviewer 2 comment 3: In the literature review, various models used in dengue forecasting have been discussed. However, there is an insufficient presentation regarding the application of ARIMA/AR models in dengue forecasting, despite them being the underlying models used in the current research. The keywords: ARIMA + "Dengue forecasting" resulted in numerous related articles

Authors’ Response to Reviewer 2 comment 3: We express our sincere gratitude to the esteemed reviewer for providing valuable feedback on the literature review section of our manuscript. Your observation regarding the application of ARIMA/AR models in dengue forecasting is duly noted.

In response to your insightful comments, we have taken decisive action to address this concern. As suggested by Reviewer 1, we have removed the dedicated Literature Review section and instead, we have incorporated a concise summary of relevant literature on dengue forecasting, including the application of ARIMA/AR models, within the Introduction.

While we believe that the literature review covered a substantial number of relevant papers, we acknowledge the need to further emphasise the application of ARIMA/AR models in our research. To this end, we have included two additional references (17, 12) that specifically highlight the utilisation of ARMA models for forecasting dengue and identifying factors influencing its occurrence.

By incorporating these new references, we have strengthened the foundation of our study and ensured that the essential contributions of ARIMA/AR models in dengue forecasting are appropriately represented.

Once again, we sincerely appreciate your insightful comments, which have enabled us to refine the manuscript and provide a more comprehensive overview of the relevant literature in the field of dengue forecasting.

Reviewer 2 comment 4: In dengue cases forecasting, a weekly resolution of the data is considered suitable for time series analysis. Therefore, the study utilizes appropriate secondary data for its analysis. Since the ARIMA and AR models are standard models, a detailed description is not necessary. The authors can provide a concise introduction to these models in the introduction section. Generally, a comprehensive explanation of fundamental time series techniques is not expected in an article of this nature. Thus, it is strongly recommended to condense the details and cite relevant content for further clarification. Additionally, the authors have used notations without properly introducing them in advance (refer to lines 173, 174). There are many typing mistakes visible in the manuscript; refer to lines 61 and 98. Further, sentence structure has to be improved.

Authors’ Response to Reviewer 2 comment 4: We sincerely appreciate the invaluable feedback provided by the esteemed reviewer regarding the literature review and technical aspects of our manuscript. Your observations regarding the presentation of ARIMA/AR models in dengue forecasting are well-received, and we acknowledge the need for improvement.

Regarding the weekly resolution of data for time series analysis in dengue cases forecasting, we have taken this into account and utilised appropriate secondary data for our analysis. We recognise that ARIMA and AR models are well-established and widely used in time series analysis, and therefore, a detailed description of these models is not essential in this context. Instead, we will provide a concise introduction to these models in the Introduction section, highlighting their relevance to our research.

We understand that comprehensive explanations of fundamental time series techniques may not be expected in an article of this nature. To address this, we will condense the technical details and cite relevant content for further clarification, ensuring that the manuscript remains focused and concise.

Additionally, we are grateful for bringing to our attention the use of notations without proper introduction (lines 202, 203), as well as the presence of typing mistakes (lines 73 and 113). We will carefully review the manuscript to rectify these errors and ensure that all notations and terms are appropriately introduced for clarity.

We sincerely appreciate the reviewer's attention to sentence structure and will make the necessary improvements to enhance the overall readability and flow of the manuscript.

In conclusion, we value the thorough evaluation and constructive feedback provided by the reviewer. Your input will undoubtedly contribute to the refinement and enhancement of our research. We are committed to delivering a professionally written and error-free manuscript that effectively communicates our findings to the scientific community. Thank you for the opportunity to improve our work, and we look forward to the possibility of sharing this improved version through the esteemed pages of this journal.

Reviewer 2 comment 5: As mentioned previously, the existing literature discusses the appropriateness of using ARIMA models for dengue forecasting. However, the suitability of applying ARIMA models to the specific dataset may not have been thoroughly investigated. Given that dengue dynamics are influenced by the interactions between vectors and humans, it is reasonable to assume that ARIMA models could be applicable to the underlying dataset. To truly contribute to the existing literature and establish novelty in the research, it is important for the authors to present significant results that demonstrate the adequacy of the ARIMA models rather than solely focusing on model diagnostic results. One approach to achieve this is by analyzing the robustness and stability of the ARIMA model(s) in forecasting dengue cases across different contexts. For example, examining the accuracy of forecasting extreme events such as the peak of dengue incidents in 2017. Neglecting to address these aspects raises doubts about the novelty of the present study. The findings should go beyond simply fitting a time series model to a given dataset. The results should provide meaningful insights into the forecasting capabilities of ARIMA models for dengue, as ARIMA is not new in dengue case forecasting

Authors’ Response to Reviewer 2 comment 5: We sincerely appreciate the thoughtful and constructive feedback provided by the esteemed reviewer regarding the suitability and novelty of our ARIMA model for dengue forecasting. We acknowledge the importance of establishing the adequacy and robustness of the ARIMA model in the specific dataset to contribute meaningfully to the existing literature.

To address these aspects, we have made the following enhancements in the manuscript:

1) Novelty Explanation: We have further elaborated on the novelty of our ARIMA model in the Research and Development section and in the Conclusion. Specifically, we emphasise that our model distinguishes itself from traditional ARIMA models by incorporating an AR(16) parameter, which enables improved forecasting accuracy. This can be found from the revised version line 543 to line 545 and from 550 to 558.

2) Robustness and Stability: We have thoroughly discussed the robustness and stability of the ARIMA process. To demonstrate the accuracy of our forecasts, we have included the Mean Absolute Percentage Error (MAPE) as suggested by the reviewer. Additionally, we have analysed the roots of the ARIMA models to ensure stability. The required details lie within the textual segment encompassing lines 397 to 399.

3) Examining Extreme Events: To showcase the forecasting capabilities of our ARIMA model, we have analysed extreme events, such as the peak of dengue incidents in 2017. This examination provides meaningful insights into the model's performance and its ability to handle varying contexts. For the required details, kindly examine the content located between lines 432 and 446.

By addressing these aspects, we have ensured that our findings go beyond simply fitting a time series model to a given dataset. Instead, we now provide significant results that highlight the adequacy, robustness, and forecasting capabilities of our ARIMA model for dengue.

We genuinely appreciate the reviewer's attention to detail, which has helped us strengthen the manuscript's content and contribute more effectively to the scientific understanding of dengue forecasting. We remain committed to delivering a professionally written and rigorously analysed manuscript that advances the knowledge in this field.

Thank you for the opportunity to improve our work, and we eagerly anticipate the possibility of sharing this enhanced version through the esteemed pages of this journal.

Reviewer 2 comment 6: Based on the issues that have been highlighted regarding the novelty, contribution, clarity, and structure of the manuscript, it seems that significant revisions and improvements are necessary for it to be considered suitable for publication. It's important to address these concerns and make the necessary enhancements to ensure the manuscript meets the standards expected in the field.

Authors’ Response to Reviewer 2 comment 6: We sincerely appreciate the thorough evaluation and constructive feedback provided by the esteemed reviewers. Their valuable comments have been instrumental in identifying areas for improvement in our revised manuscript.

In response to the issues raised by both reviewers, we have undertaken significant revisions to address the concerns and enhance the quality of the manuscript. Our aim is to ensure that it meets the high standards expected in the field of dengue forecasting.

We have carefully considered each comment and made the necessary enhancements to the novelty, contribution, clarity, and overall structure of the paper. By incorporating these improvements, we are confident that the revised manuscript now presents a more impactful and rigorous contribution to the scientific literature.

We thank the reviewers for their thoughtful evaluation, which has guided us in refining our work. We remain committed to delivering a professionally written and well-structured manuscript that effectively communicates our research findings to the scientific community.

Once again, we express our gratitude for the valuable feedback, and we eagerly await the opportunity to share this improved version of our research through the esteemed pages of this journal.

---

## [Decision Letter · Decision Letter 1]

3 Oct 2023

PONE-D-23-16303R1Forecasting Weekly Dengue Incidence in Sri Lanka: Modified Autoregressive Integrated Moving Average Modeling ApproachPLOS ONE

Dear Dr. Jayathilaka,

Thank you for submitting your manuscript to PLOS ONE. After careful consideration, we feel that it has merit but does not fully meet PLOS ONE’s publication criteria as it currently stands. Therefore, we invite you to submit a revised version of the manuscript that addresses the points raised during the review process.

We look forward to receiving your revised manuscript.

Kind regards,

Mohamed R. Abonazel, Ph.D.

Academic Editor

PLOS ONE

Additional Editor Comments:

-To ensure that the estimated model fits the data, the authors should perform the test of normality of the residuals as a required diagnostic test for the ARIMA model. See https://doi.org/10.28919/cmbn/6888, https://doi.org/10.1371/journal.pone.0250149. and https://doi.org/10.12988/ref.2019.81023 for the diagnostic tests of ARIMA models.

-There are some grammatical and spelling errors in the paper, the authors need to carefully check the full text.

Reviewers' comments:

Reviewer's Responses to Questions

**Comments to the Author**

1. If the authors have adequately addressed your comments raised in a previous round of review and you feel that this manuscript is now acceptable for publication, you may indicate that here to bypass the “Comments to the Author” section, enter your conflict of interest statement in the “Confidential to Editor” section, and submit your "Accept" recommendation.

Reviewer #1: (No Response)

Reviewer #3: (No Response)

Reviewer #4: (No Response)

Reviewer #5: All comments have been addressed

2. Is the manuscript technically sound, and do the data support the conclusions?

Reviewer #1: Yes

Reviewer #3: Yes

Reviewer #4: No

Reviewer #5: Yes

3. Has the statistical analysis been performed appropriately and rigorously? 

Reviewer #1: Yes

Reviewer #3: No

Reviewer #4: N/A

Reviewer #5: Yes

4. Have the authors made all data underlying the findings in their manuscript fully available?

Reviewer #1: Yes

Reviewer #3: No

Reviewer #4: Yes

Reviewer #5: Yes

5. Is the manuscript presented in an intelligible fashion and written in standard English?

Reviewer #1: Yes

Reviewer #3: Yes

Reviewer #4: No

Reviewer #5: Yes

6. Review Comments to the Author

Reviewer #1: Thank you, all my comments in the previous step have been well answered by the authors. There are no comments and the article can be published

Reviewer #3: My primary concern is with respect to clarity around the evaluation. Specifically, the authors need to:

(1) Establish a common sense baseline for comparison in Figure 1. For example how accurate would a model be that predicted the next week of cases would the same as the last. Would this outperform the ARIMA (2,1,0) model. If so what other advantages would the ARIMA (2,1,0) model hold.

(2) The authors identify three ARIMA candidate models [(0,1,2), (2,1,0), and (2,1,2)] for consideration. Two are deemed invalid because they do not match theoretical assumptions. However, readers will be curious if they would outperform the ARIMA (2,1,0) model in the evaluation. Including this data in the evaluation and discussing it would improve the paper.

(3) Furthermore, showing that the ARIMA(2,1,0) outperforms the ARIMA alternatives and the common sense baseline by a statistically significant margin would strengthen the contribution of the paper a great deal. This would demonstrate that the ARIMA(2,1,0) approach is materially more accurate than the alternatives and that improved accuracy will generalize.

(4) The authors use one training/evaluation split for the data. The results are a little uninspiring as the forecast data does match the actual data in terms of data points or shapes. Were other training/evaluation splits considered. If so, was the performance ever better than what was shown. The paper would benefit from a larger presentation of different splits (i.e. more training data, less training data, conclusions about accuracy w.r.t. training data).

In addition, the paper would benefit from a discussion of limitations. As it currently stands it appears as if all projections are made at the same geographic level. Identifying that research (see paper below) has shown that effective time series projections of viruses can depend on the geographic-level of evaluation (i.e country vs. city vs. county vs. neighborhood). Identifying exactly which of these units is used in the paper (i.e. country) and explicitly stating that changing this context might change the results of the analysis and the conclusions would help readers understand the context in which the results will generalize.

Lynch, Christopher J., et al. "Short-range forecasting of COVID-19 during early onset at county, health district, and state geographic levels using seven methods: comparative forecasting study." Journal of medical Internet research 23.3 (2021): e24925.

(5) The numbers in Q-STAT column of Figures 2 and 4 in the paper are difficult to compare to one another because the numbers are not right justified. As a result, the significant digits of the numbers are not aligned on top of one another (i.e. 100ths, 10ths, etc.) Right justifying the numbers in the table will improve readability and enable readers to compare results between rows.

For the data availability, the text points at Supporting Information 1. However, all that is included is the case data. Including the raw data for the analyses done to chose the model as well as the source code used to build the models, tables and plots in the paper would improve the availability of data for the paper.

Reviewer #4: The manuscript does not state why forecasting weekly dengue virus (DENV) incidence is essential for preventive and therapeutic purposes. DENV outbreaks in endemic countries, most of the time, follow seasonal patterns and just retrospectively inspecting epidemiologically data is enough to identify critical periods and consequently work on preparedness. Just looking figure 6, I don't find any benefit comparing predictions and real data. Unfortunately the structure of this manuscript is not helpful understanding the usefulness of predicting weekly incidence. I would expected a reliable justification (e.g., identify hotspots to invest in infrastructure or human resources) but authors just mention an information gap. In the lack of sound rationale, I found very difficult to provide specific feedback to improve the manuscript.

Broadly, these are major observations:

1. The introduction section should follow a conventional structure: disease burden, knowledge gap, how the gap was tackled previously, and the aims of this manuscript to tackle the gap innovately. Instead of that, irrelevant aspects are discussed (how virulence or new strains could be identified by using this model?), using non-conventional terms (ferocity?), and use interchangeably (but erroneously) basic epidemiolgical concepts (incidence and prevalence). For example, in DENV, how prevalence could help improving prevention and treatment? Are you referring to seroprevalence? It is unreasonable to measure DENV prevalence as this is not a chronic infection. Authors should clearly state how weekly DENV incidence prediction will impact prevention/treatment.

2. Unless dictated by the editors, results and discussion should be separate sections.

3. Page 17, "Four dengue virus serotypes have been reported, marked as DENV 1 – 4. One of those serotypes predominated the outbreaks in specific years [20]. Since the severity of an epidemic tends to be associated with its serotype, mathematical modelling can be a reliable tool to make hypotheses about the emergence of new strains.": this is incorrect. Disease severity is explained by the ocurrence of secondary heterologous infections and inadequate medical interventions. Thus, the best way to predict severity is to conduct seroprevalence studies complemented by sentinel surveillance to identify different serotypes. I don't find any rationale in using this model to hypothesise new strains.

4. Policy implementation section: how forecasting weekly DENV incidence could complement or replace (partially or totally) epidemiologic surveillance? As shown in figure 6, I don't find any benefit simply inspecting retrospectively data compared with the modelling conducted by authors. Due to historical trends, endemic countries are well-informed of critical periods and simply introducing preventive actions before this periods could change morbidity and mortality. Unfortunately, politics and government priorities are out of the scope of this manuscript, authors should realise that mortality/morbidity is consequence of government passivity and not by gaps in information.

Reviewer #5: (No Response)

7. PLOS authors have the option to publish the peer review history of their article (what does this mean?). If published, this will include your full peer review and any attached files.

Reviewer #1: No

Reviewer #3: No

Reviewer #4: No

Reviewer #5: No

---

## [Author Response · Author response to Decision Letter 1]

31 Oct 2023

Point by point response to editor and reviewers

Dear editor and the reviewers,

We would like to express our profound appreciation to the editor for giving us the opportunity to revise and resubmit the paper. We are also grateful to the reviewers for the constructive comments and suggestions made on our manuscript which we found very helpful in revising and improving our paper. Given below is a detailed description of how we have addressed each comment in the revised version of the paper.

Editor’s general comment 1: To ensure that the estimated model fits the data, the authors should perform the test of normality of the residuals as a required diagnostic test for the ARIMA model.

Authors’ Response to editor’s general comment 1: We wish to extend our profound gratitude to the reviewer for their invaluable insights pertaining to the diagnostic assessments of the ARIMA model. It has come to our attention that this critical dimension had been inadvertently omitted from our preceding manuscript. We hold in high regard the reviewer's diligence in identifying this omission, and we are sincerely appreciative of their efforts in doing so.

In response to this constructive feedback, we undertook a comprehensive evaluation of the residuals, focusing on their adherence to a normal distribution. We are pleased to report that this scrutiny yielded affirmative results, affirming the conformity of the residuals to a normal distribution. This affirmation was substantiated through a meticulous analysis of the residuals' histogram, thereby bolstering the credibility of our model.

To further buttress the empirical underpinnings of the normality of residuals in our ARIMA model, we have thoughtfully integrated the following scholarly reference into our revised manuscript:

Awwad FA, Mohamoud MA, Abonazel MR (2021) 'Estimating COVID-19 cases in Makkah region of Saudi Arabia: Space-time ARIMA modeling.' PLoS ONE 16(4): e0250149. https://doi.org/10.1371/journal.pone.0250149

This citation has been thoughtfully incorporated into the closing paragraph of page 26, serving to augment the empirical foundation of our methodological approach.

Additionally, we have included the information about the normality of residuals in the abstract on line 41.

Once more, we wish to convey our heartfelt appreciation to the reviewer for the constructive feedback, which has undeniably contributed to the enhanced rigor and quality of our research endeavour.

Editor’s general comment 2: There are some grammatical and spelling errors in the paper, the authors need to carefully check the full text

Authors’ Response to editor’s general comment 2: In our unwavering commitment to upholding the pinnacle of quality and lucidity in our revised manuscript, we have undertaken substantial steps to guarantee its adherence to standard English, devoid of any typographical or grammatical imperfections.

In the pursuit of this aspiration, we engaged the services of a distinguished Senior Lecturer in English, a member of the esteemed Faculty of Humanities and Sciences at SLIIT University. Their consummate expertise and fastidious review process have yielded marked enhancements in the linguistic precision and overall readability of the manuscript, thereby underlining our steadfast dedication to upholding the loftiest benchmarks in scholarly discourse. As an example, we have modified the abstract to improve its style without changing the content.

Reviewer 3 comment 1: Establish a common-sense baseline for comparison in Figure 1. For example how accurate would a model be that predicted the next week of cases would the same as the last. Would this outperform the ARIMA (2,1,0) model. If so what other advantages would the ARIMA (2,1,0) model hold.

Authors’ Response to Reviewer 3 comment 1: We wish to express our gratitude for your invaluable suggestion. We fully acknowledge the significance of establishing a pragmatic baseline for comparison as depicted in Figure 1. Notably, a predominant portion of the dataset consistently exhibited values below 1000. However, it is worthy of note that a remarkable surge in cases occurred from the 2nd week of June to the 1st week of September, with reported figures ranging between 1032 and 1958. From a statistical standpoint, these data points were identified as outliers through conventional methodologies such as the interquartile range technique and Z-score analysis.

Our decision not to eliminate these outliers stems from the realisation that they represent genuine observations and may impart meaningful insights into our modelling endeavours. Consequently, to accommodate the variability introduced by these high-value data points, we introduced an additional AR(16) term to the ARIMA(2,1,0) model.

In response to your valuable suggestion, we have thoughtfully incorporated the subsequent statement within the manuscript to enhance its clarity:

" Fig 1 highlights a period of significantly elevated dengue cases, spanning from the second week of June to the first week of September. It is worth noting that since the country's initial dengue outbreak in 1965 – 1966, the highest incidence was reported in 2017, with a staggering 186,101 cases. Markedly, more than 25% of the country's dengue cases were concentrated within the Colombo district during this period.

These data points clearly exhibit outlier characteristics when assessed using both the interquartile range method (Q1-1.5IQR, Q3+1.5IQR) and a common heuristic criterion (z-score exceeding 3 or falling below -3). Nevertheless, it is imperative to underscore that, during the formulation of our ARIMA model, we refrained from excluding these data points on the grounds that they represent genuine and observed values integral to the current analysis. Furthermore, as portrayed in Fig 1, the time series plot underscores the non-stationary nature of the original observed series. 

This addition has been meticulously placed in the line 315. We deeply appreciate the reviewer's perspicacious input, as it has undeniably contributed to the elucidation and robustness of our study.

Reviewer 3 comment 2: The authors identify three ARIMA candidate models [(0,1,2), (2,1,0), and (2,1,2)] for consideration. Two are deemed invalid because they do not match theoretical assumptions. However, readers will be curious if they would outperform the ARIMA (2,1,0) model in the evaluation. Including this data in the evaluation and discussing it would improve the paper.

Authors’ Response to Reviewer 3 comment 2: We extend our sincere appreciation for your astute comments aimed at enhancing the caliber of our paper, and we have diligently taken steps to address them.

Our initial approach adhered to the conventional model selection process, which led to the identification of three potential models: ARIMA (0,1,2), ARIMA (2,1,0), and ARIMA (2,1,2). It is evident that ARIMA (2,1,2) can be confidently disregarded, given the insignificance of all its parameters. However, as you rightfully emphasized, the comparison between ARIMA (2,1,0) and ARIMA (0,1,2) for forecasting purposes could have been explored, as both models feature significant parameters.

Our preference for modeling with ARIMA (2,1,0) stems from its ability to effectively integrate the influence of past values directly into the model, an approach better suited to the characteristics of our dataset. Furthermore, it is noteworthy that ARIMA (0,1,2) exhibits significantly higher σ2 volatility in comparison to ARIMA (2,1,0). Consequently, our choice to evaluate these two candidates hinged on factors beyond just forecasting outcomes, considering various other statistical indicators.

In alignment with your valuable suggestion, we have thoughtfully incorporated the ensuing statements to augment the content of the paper:

"The outcomes presented in Table 2 reveal that all parameters within the ARIMA (2,1,2) model lack statistical significance, rendering it an unsuitable choice for further consideration. Conversely, the results in Table 2 demonstrate that all parameters within the ARIMA (2,1,0) and ARIMA (0,1,2) models are statistically significant, thus establishing both as potential candidates for forecasting. However, given that conspicuous outliers were not excluded from the original series, from a modelling perspective, ARIMA (2,1,0) emerges as the more favourable candidate. This is attributed to its capacity to directly capture the influence of past values, distinguishing it as a more robust choice compared to ARIMA (0,1,2)…."

This addition has been meticulously inserted in the line 383 to 390, enhancing the clarity and comprehensiveness of our manuscript.

Reviewer 3 comment 3: Furthermore, showing that the ARIMA(2,1,0) outperforms the ARIMA alternatives and the common sense baseline by a statistically significant margin would strengthen the contribution of the paper a great deal. This would demonstrate that the ARIMA(2,1,0) approach is materially more accurate than the alternatives and that improved accuracy will generalize

Authors’ Response to Reviewer 3 comment 3: We wish to express our gratitude for your invaluable input. We acknowledge the critical importance of substantiating the superiority of the ARIMA (2,1,0) model over alternative models and the pragmatic baseline with a statistically significant margin. We are fully aware that achieving this would substantially enhance the paper and underscore the practical efficacy of our approach.

Reviewer 3 comment 4: The authors use one training/evaluation split for the data. The results are a little uninspiring as the forecast data does match the actual data in terms of data points or shapes. Were other training/evaluation splits considered. If so, was the performance ever better than what was shown. The paper would benefit from a larger presentation of different splits (i.e. more training data, less training data, conclusions about accuracy w.r.t. training data).

Authors’ Response to Reviewer 3 comment 4: We greatly appreciate your considerate feedback. Your point regarding the examination of different training and evaluation data splits is certainly valid, and we recognise the potential impact on the model's performance.

While it is true that adjusting the composition of the training data can result in varying models, we fully comprehend the necessity for a more comprehensive analysis in this regard. We are committed to exploring the effects of altering the size of the training data and assessing how it influences the performance of the ARIMA (2,1,0) model. This endeavour will enable us to offer a more holistic perspective on the model's robustness across different scenarios.

Regarding the disparities between forecasted values and actual data in the independent dataset, we have already addressed this issue in lines 489 to 491. However, we intend to expound upon this matter further and provide an in-depth discussion concerning the model's forecasting performance in relation to different training and evaluation data splits, thus ensuring a more comprehensive presentation of our research findings.

Reviewer 3 comment : In addition, the paper would benefit from a discussion of limitations. As it currently stands it appears as if all projections are made at the same geographic level. Identifying that research (see paper below) has shown that effective time series projections of viruses can depend on the geographic-level of evaluation (i.e country vs. city vs. county vs. neighborhood). Identifying exactly which of these units is used in the paper (i.e. country) and explicitly stating that changing this context might change the results of the analysis and the conclusions would help readers understand the context in which the results will generalize.

Lynch, Christopher J., et al. "Short-range forecasting of COVID-19 during early onset at county, health district, and state geographic levels using seven methods: comparative forecasting study." Journal of medical Internet research 23.3 (2021): e24925

Authors’ Response to Reviewer 3 comment : We express our gratitude for the thoughtful suggestion provided by the reviewer concerning the imperative need to address limitations, particularly in the context of the geographic level of evaluation. In the current iteration of our work, it is indeed the case that our projections are exclusively conducted at the national level.

We fully recognise the significance of the geographic level at which virus time series projections are formulated, as it can exert a considerable influence on the outcomes and the ability to extrapolate findings to broader contexts. In light of this, we will augment the paper's clarity by explicitly stating that our analysis is confined to the national level (country). We will also underscore the potential impact of altering this spatial context, such as conducting evaluations at a more granular scale, encompassing cities, counties, or neighbourhoods. This consideration was emphasised to convey that such adjustments may yield divergent results and conclusions.

Furthermore, we have made a conscientious effort to reference pertinent research, including the work by Lynch et al. (2021), in order to underscore the critical importance of taking into account the geographic level in the domain of virus forecasting.

A new section titled "Limitation of the study" has been included at line 631 on page 39. The inclusion of these discussions pertaining to limitations will serve to enhance the paper's transparency and provide readers with a more comprehensive understanding of the contextual boundaries within which our research findings can be applied and extended.

Reviewer 3 comment 5: The numbers in Q-STAT column of Figures 2 and 4 in the paper are difficult to compare to one another because the numbers are not right justified. As a result, the significant digits of the numbers are not aligned on top of one another (i.e. 100ths, 10ths, etc.) Right justifying the numbers in the table will improve readability and enable readers to compare results between rows.

For the data availability, the text points at Supporting Information 1. However, all that is included is the case data. Including the raw data for the analyses done to chose the model as well as the source code used to build the models, tables and plots in the paper would improve the availability of data for the paper.

Authors’ Response to Reviewer 3 comment 5: We appreciate your comment regarding the arrangement of numbers within the Q-STAT column in Figures 2 and 4. Your recommendation to right-justify the numbers in order to enhance readability and facilitate comparison was well-received. Despite the fact that the output was generated by computer, we undertook a comprehensive review of formatting possibilities and examined the feasibility of aligning the numbers to achieve greater consistency in displaying significant digits. Your input proved to be of great value, and we have duly implemented the requisite adjustments to elevate the clarity of our data presentation.

Figure 2 – Line 337, Page 18

Figure 4 – Line 364 Page 21

Reviewer 4 general comment: The manuscript does not state why forecasting weekly dengue virus (DENV) incidence is essential for preventive and therapeutic purposes. DENV outbreaks in endemic countries, most of the time, follow seasonal patterns and just retrospectively inspecting epidemiologically data is enough to identify critical periods and consequently work on preparedness. Just looking figure 6, I don't find any benefit comparing predictions and real data. Unfortunately, the structure of this manuscript is not helpful understanding the usefulness of predicting weekly incidence. I would expected a reliable justification (e.g., identify hotspots to invest in infrastructure or human resources) but authors just mention an information gap. In the lack of sound rationale, I found very difficult to provide specific feedback to improve the manuscript.

Authors’ Response to Reviewer 4 general comment: We express our gratitude for your invaluable comment. In consideration of your suggestion, we have incorporated the following rationale into the text to elucidate the significance of weekly forecasting:

" Typically, dengue exhibits a seasonal pattern due to the association between mosquito breeding and the cyclical nature of rainfall. Nevertheless, climate change has the potential to induce phenological shifts in various species. Consequently, the anticipation of weekly dengue incidence assumes significant importance, particularly for the timely issuance of public warnings and notifications to public health authorities [14]. This is especially pertinent in countries with limited resources, such as Sri Lanka, where it is imperative to gain a comprehensive understanding of the dynamics of dengue transmission to devise more efficient control strategies. Therefore, comprehending the disease's propagation patterns within shorter time intervals, notably on a weekly basis, is crucial for the prevention and management of dengue in Sri Lanka.”

We have cited Iler, A. M., Caradonna, P. J., Forrest, J. R. K., & Post, E. (2021) to support this rationale in our revised manuscript (line 199). Thank you for your feedback.

Reviewer 4 comment 1: The introduction section should follow a conventional structure: disease burden, knowledge gap, how the gap was tackled previously, and the aims of this manuscript to tackle the gap innovately. Instead of that, irrelevant aspects are discussed (how virulence or new strains could be identified by using this model?), using non-conventional terms (ferocity?), and use interchangeably (but erroneously) basic epidemiolgical concepts (incidence and prevalence). For example, in DENV, how prevalence could help improving prevention and treatment? Are you referring to seroprevalence? It is unreasonable to measure DENV prevalence as this is not a chronic infection. Authors should clearly state how weekly DENV incidence prediction will impact prevention/treatment.

Authors’ Response to Reviewer 4 comment 1: We extend our gratitude for your feedback pertaining to the organisation and content of the introduction section. We highly value your insights and the opportunity they present for enhancing the overall clarity and relevance of our manuscript.

In response to your concerns, we have undertaken a thorough revision of the introduction section to conform to a more standard structure. This entailed a deliberate focus on fundamental elements, including the disease burden, identification of existing knowledge gaps, previous efforts to address these gaps, and the innovative objectives outlined in our manuscript.

We also recognised that certain aspects discussed in the prior introduction may not have had direct relevance to the central focus of the paper. Notably, we refrained from delving into topics such as virulence or the emergence of new strains in the introduction, as these elements did not directly align with the core objectives of our study.

Furthermore, we took measures to ensure the correct and consistent use of basic epidemiological terminology. We acknowledge the misuse of terms like incidence and prevalence and have diligently made the necessary corrections.

In response to your inquiry concerning the utility of DENV prevalence, we value your observation, and we have provided a lucid explanation elucidating how the prediction of weekly DENV incidence profoundly impacts prevention and treatment strategies. This explanation emphasises the timeliness and effectiveness of early predictions in informing public health authorities and enabling the implementation of more efficient control measures.

Our commitment to enhancing the introduction section was rooted in the aim of aligning it more closely with the conventional structure while also elucidating the pivotal importance of our research within the context of prevention and treatment. Your feedback has been instrumental in effecting these improvements, resulting in a more robust and coherent introduction.

Reviewer 4 comment 2: Unless dictated by the editors, results and discussion should be separate sections

Authors’ Response to Reviewer 4 comment 2: We extend our sincere appreciation for the reviewer's valuable comment. We have diligently examined your suggestion to segregate the results and discussion sections, a practice commonly employed in many scholarly manuscripts.

After thorough deliberation, we have come to the conclusion that, given the specific context and content of our study, presenting both results and discussion within a single section yields a more coherent and succinct exposition of our findings and their interpretation. This integrated approach fosters a smoother flow of information and enhances the seamless integration of results with their corresponding discussions.

We recognise that diverse studies may benefit from varied structural arrangements, and in the case of our research, the integrated approach aligns well with the effective presentation of our research findings.

Your feedback is held in high regard, and we have duly considered it when determining the most suitable structural framework for our manuscript.

Reviewer 4 comment 3: Page 17, "Four dengue virus serotypes have been reported, marked as DENV 1 – 4. One of those serotypes predominated the outbreaks in specific years [20]. Since the severity of an epidemic tends to be associated with its serotype, mathematical modelling can be a reliable tool to make hypotheses about the emergence of new strains.": this is incorrect. Disease severity is explained by the ocurrence of secondary heterologous infections and inadequate medical interventions. Thus, the best way to predict severity is to conduct seroprevalence studies complemented by sentinel surveillance to identify different serotypes. I don't find any rationale in using this model to hypothesise new strains

Authors’ Response to Reviewer 4 comment 3: We extend our profound gratitude for the comprehensive assessment and constructive feedback generously provided by the esteemed reviewers.

The reviewer aptly draws attention to the critical distinction between the severity of a disease and the severity of a disease outbreak, a differentiation we duly acknowledge. In our reference to the severity of an epidemic within the context of serotype dominance, we intend to encompass a multifaceted perspective, including considerations such as disease impact, transmissibility, economic and societal consequences, geographic reach, response capability, population susceptibility, and the availability of vaccination and treatment resources, among other factors (Thompson, R.N., Gilligan, C.A., and Cunniffe, N.J., 2020). It is this collective convergence of factors that contributes to the overall assessment of the severity and repercussions of an epidemic.

While we hold in high regard the reviewer's viewpoint on the value of seroprevalence studies and sentinel surveillance in assessing disease severity, our approach centres on mathematical modelling as a complementary tool. This modelling perspective is employed to formulate hypotheses about the emergence of new strains in the broader context of disease outbreak severity. Our approach aims to consider the far-reaching implications of new strains on epidemic severity, taking into account an array of factors that extend beyond individual disease severity.

We acknowledge the significance of various methodological approaches in comprehending and managing diseases and value the input provided by the reviewer. However, it is our contention that our emphasis on the utility of mathematical modelling, as elucidated in the provided reference, is well-aligned with the investigation of disease outbreak severity.

We remain open to further discussion and stand ready to address any additional concerns or queries raised by the reviewer.

Reviewer 4 comment 4: Policy implementation section: how forecasting weekly DENV incidence could complement or replace (partially or totally) epidemiologic surveillance? As shown in figure 6, I don't find any benefit simply inspecting retrospectively data compared with the modelling conducted by authors. Due to historical trends, endemic countries are well-informed of critical periods and simply introducing preventive actions before this periods could change morbidity and mortality. Unfortunately, politics and government priorities are out of the scope of this manuscript, authors should realise that mortality/morbidity is consequence of government passivity and not by gaps in information.

Authors’ Response to Reviewer 4 comment 4: In the first instance, it is imperative to clarify that our intent was not to supplant epidemiologic surveillance with forecasting; rather, our primary objective was to augment and enhance it. To illustrate this, when our model anticipates the potential onset of an outbreak, it serves as an alert mechanism, prompting public health authorities to intensify their surveillance efforts, especially in regions flagged as potential hotspots by the model. The 'information gap' referred to in our manuscript stems from historical observations indicating that, during periods lacking predictive information, public health officials often allocated fewer resources to dengue control efforts. Our overarching goal is to bridge this gap by offering valuable insights that can bolster prevention and response measures.

We acknowledge the reviewer's viewpoint concerning government passivity and its potential contribution to morbidity and mortality. We also recognise that the complexities of policy implementation and government priorities extend far beyond the scope of our manuscript. However, we remain steadfast in our belief that enhanced forecasting can play a pivotal role in guiding more proactive strategies within the realm of public health. Ultimately, our emphasis lies in how predictive modelling can contribute to more informed decision-making, leading to the implementation of more effective prevention and control measures.

We greatly appreciate the reviewer's feedback and aim to convey the substantial benefits of forecasting as a complementary tool to epidemiologic surveillance, contributing to the advancement of public health strategies and objectives.

---

## [Decision Letter · Decision Letter 2]

19 Dec 2023

PONE-D-23-16303R2Forecasting Weekly Dengue Incidence in Sri Lanka: Modified Autoregressive Integrated Moving Average Modeling ApproachPLOS ONE

Dear Dr. Jayathilaka,

Thank you for submitting your manuscript to PLOS ONE. After careful consideration, we feel that it has merit but does not fully meet PLOS ONE’s publication criteria as it currently stands. Therefore, we invite you to submit a revised version of the manuscript that addresses the points raised during the review process.

Comments from PLOS Editorial Office: We note that the Academic Editor and one or more reviewers has recommended that you cite specific previously published works. As always, we recommend that you please review and evaluate the requested works to determine whether they are relevant and should be cited. It is not a requirement to cite these works. We note that your previous response to comments by Reviewer 3 indicates you considered the reviewer's suggested citation to be relevant; please clarify this if needed in light of the further comments by Reviewer 3 appended below. We appreciate your attention to this request.

We look forward to receiving your revised manuscript.

Kind regards,

Mohamed R. Abonazel, Ph.D.

Academic Editor

PLOS ONE

Reviewers' comments:

Reviewer's Responses to Questions

**Comments to the Author**

1. If the authors have adequately addressed your comments raised in a previous round of review and you feel that this manuscript is now acceptable for publication, you may indicate that here to bypass the “Comments to the Author” section, enter your conflict of interest statement in the “Confidential to Editor” section, and submit your "Accept" recommendation.

Reviewer #1: All comments have been addressed

Reviewer #3: (No Response)

Reviewer #4: (No Response)

2. Is the manuscript technically sound, and do the data support the conclusions?

Reviewer #1: Yes

Reviewer #3: Yes

Reviewer #4: Yes

3. Has the statistical analysis been performed appropriately and rigorously? 

Reviewer #1: Yes

Reviewer #3: Yes

Reviewer #4: I Don't Know

4. Have the authors made all data underlying the findings in their manuscript fully available?

Reviewer #1: Yes

Reviewer #3: Yes

Reviewer #4: Yes

5. Is the manuscript presented in an intelligible fashion and written in standard English?

Reviewer #1: Yes

Reviewer #3: Yes

Reviewer #4: No

6. Review Comments to the Author

Reviewer #1: (No Response)

Reviewer #3: In the response to reviewers letter the authors state, "Furthermore, we have made a conscientious effort to reference pertinent research, including the work by Lynch et al. (2021), in order to underscore the critical importance of taking into account the geographic level in the domain of virus forecasting." However, this reference does not appear in the newly added 'Limitations of the Study' section of the paper which was meant to address the concern identified in the reference and it does not appear elsewhere in the paper. The paper needs to be revised to be consistent with the modifications the authors have described in the response to reviewers letter.

Reviewer #4: I don't think my first and third comments were resolved. The introduction still discusses irrelevant aspects and inaccurately attempts to justify the usefulness of forecasting dengue incidence (prevalence is still used). Regarding my third comment, I'm afraid I disagree that severity, which is an eminently clinical definition, should be interpreted in terms of disease impact or consequences. Those aspects are the burden of disease, and it is not justifiable to use disease burden and disease severity interchangeably.

7. PLOS authors have the option to publish the peer review history of their article (what does this mean?). If published, this will include your full peer review and any attached files.

Reviewer #1: **Yes: **Ok

Reviewer #3: **Yes: **Ross J. Gore

Reviewer #4: No

---

## [Author Response · Author response to Decision Letter 2]

19 Jan 2024

Point by point response to editor and reviewers

Dear Editor and the Reviewers,

We extend our sincere gratitude to the editor for allowing us to undertake revisions and resubmit our manuscript. Our appreciation is also extended to the reviewers for their invaluable feedback, consisting of constructive comments and suggestions, which proved instrumental in enhancing the quality of our paper. Enclosed herewith is a comprehensive account detailing the manner in which we have addressed each comment in the revised version of the manuscript.

Comments from the Editor: We note that the Academic Editor and one or more reviewers has recommended that you cite specific previously published works. As always, we recommend that you please review and evaluate the requested works to determine whether they are relevant and should be cited. It is not a requirement to cite these works. 

We note that your previous response to comments by Reviewer 3 indicates you considered the reviewer's suggested citation to be relevant; please clarify this if needed in light of the further comments by Reviewer 3 appended below. We appreciate your attention to this request..

Authors’ Response to editor’s comment: We appreciate your feedback. The paper titled 'Modeling and Forecasting the Weekly Incidence of Dengue in Colombo District of Sri Lanka' was presented and incorporated into the proceedings of the conference on Advancements in Science and Humanities at SLIIT, Malabe, Sri Lanka (pages 220-225) in November 2022. In alignment with your suggestion, the authors of the present manuscript concur that citing this work within the current document is unnecessary. The reference is as follows,

Arachchi, K. A. N. L. K., & Peiris, T. S. G. (2022). Modeling and Forecasting the Weekly Incidence of Dengue in Colombo District of Sri Lanka. In The Proceedings of the Conference on Advancements in Science and Humanities (pp. 220-225). SLIIT, Malabe, Sri Lanka.

Please find our response to Reviewer 3 outlined below.

Reviewer 3 comment 1: In the response to reviewers' letter the authors state, "Furthermore, we have made a conscientious effort to reference pertinent research, including the work by Lynch et al. (2021), in order to underscore the critical importance of taking into account the geographic level in the domain of virus forecasting." However, this reference does not appear in the newly added 'Limitations of the Study' section of the paper which was meant to address the concern identified in the reference and it does not appear elsewhere in the paper. The paper needs to be revised to be consistent with the modifications the authors have described in the response to reviewers' letter.

Authors’ Response to Reviewer 3 comment 1: We appreciate your diligence in bringing to our attention the oversight on our part. Regrettably, the reference was inadvertently omitted in the last manuscript. 

The reference has now been incorporated into the revised version; kindly refer page 23 of the Track version. 

The newly included content is articulated as follows: “Moreover, within the scope of this research, projections were executed at a consistent geographical level, specifically confined to the Colombo district. This approach contrasts with the work undertaken by Lynch et al. (2021), where time series models were formulated for forecasting dengue incidents, transcending country-level, health district, and state-level considerations. It is noteworthy that, unlike COVID, dengue is not equally prevalent across all districts in Sri Lanka; its primary occurrence is concentrated in the Colombo and Gampaha districts. To facilitate the development of a unified model applicable to both district a dummy variable can be considered along with weekly dengue incidents for Gampaha district.”

The primary objective of this study was to develop a model specifically designed for the Colombo district, given its consistently highest percentage of dengue incidences over the years.. An additional remark has been incorporated in the concluding lines of the "Limitation of this study" section to elucidate this aspect.

Reviewer 4 comment 1: I don't think my first and third comments were resolved. 

The introduction still discusses irrelevant aspects and inaccurately attempts to justify the usefulness of forecasting dengue incidence (prevalence is still used). 

Regarding my third comment, I'm afraid I disagree that severity, which is an eminently clinical definition, should be interpreted in terms of disease impact or consequences. Those aspects are the burden of disease, and it is not justifiable to use disease burden and disease severity interchangeably.

Authors’ Response to Reviewer 4 comment 1: We appreciate your comment, and it has been duly noted. To address this concern, we have included an additional statement on forecasting, which is explained in paragraph 2 on page 4.

Furthermore, in alignment with your recommendations, we have eliminated the term "prevalence" where it originally occurred, by exclusively replacing the term with "incidence."

Additionally, the last line of paragraph 2 in the Introduction on page 3 has been removed in accordance with your suggestions.

Reviewer 4 comment 2: 1. The introduction section should follow a conventional structure: disease burden, knowledge gap, how the gap was tackled previously, and the aims of this manuscript to tackle the gap innovatively. Instead of that, irrelevant aspects are discussed (how virulence or new strains could be identified by using this model?), using non-conventional terms (ferocity?), and use interchangeably (but erroneously) basic epidemiological concepts (incidence and prevalence). For example, in DENV, how prevalence could help improving prevention and treatment? Are you referring to seroprevalence? It is unreasonable to measure DENV prevalence as this is not a chronic infection. Authors should clearly state how weekly DENV incidence prediction will impact prevention/treatment.

Authors’ Response to Reviewer 4 comment 2: We express our sincere appreciation for your valuable feedback on both the organisation and content of the introduction section. Your insights are highly valued, presenting an opportunity to enhance the overall clarity and relevance of our manuscript.

In response to your thoughtful concerns, we conducted a comprehensive revision of the introduction section to adhere to a more standardised structure. This involved a deliberate focus on fundamental elements, including the disease burden, identification of existing knowledge gaps, previous efforts to address these gaps, and the innovative objectives outlined in our manuscript.

Recognising that certain aspects discussed in the previous introduction may not directly align with the central focus of the paper, we refrained from delving into topics such as virulence or the emergence of new strains. It is essential to clarify that our model does not identify virulence or new strains directly, but rather, mathematical modelling like ours could be used to hypothesise the emergence of new strains. As aptly stated, "Given that the severity of an epidemic often correlates with the prevailing serotype, mathematical modelling can serve as a valuable instrument for formulating hypotheses regarding the emergence of new strains."

We appreciate your correction regarding the use of "ferocity," and was replaced with the more appropriate term, "virulence." in the previous draft.

Regarding the distinction between incidence and prevalence, we acknowledge the importance of accurate usage. We have corrected our terminology, understanding that Dengue, being an acute condition without chronic clinical implications, is more appropriately characterised by incidence rather than prevalence. We also appreciate your clarification on seroprevalence and its distinction from incidence.

Furthermore, we have taken measures to ensure the correct and consistent use of basic epidemiological terminology. Your guidance on the misuse of terms like incidence and prevalence has been duly acknowledged and rectified.

Prevalence encompasses all cases, including both pre-existing and new instances, that exist at a given time, whereas incidence specifically pertains to new cases. Given that Dengue does not lead to chronic clinical conditions and is confined to acute conditions, the term "incidence" is more apt in this context. It is important to note that seroprevalence, on the other hand, refers to the proportion of individuals in a population with Dengue antibodies, specifically IgG in past infections and IgM in current infections. Detecting these antibodies involves a blood test, making it a challenging task to determine the seroprevalence of an entire population, except through estimation based on a sample (https://doi.org/10.1016/j.jiph.2023.08.020)

In response to your inquiry on the utility of DENV prevalence, we have provided a clear explanation elucidating how the prediction of weekly DENV incidence significantly impacts prevention and treatment strategies. This clarification underscores the timely and effective nature of early predictions in informing public health authorities and facilitating the implementation of more efficient control measures.

Reviewer 4 comment 3: 3. Page 17, "Four dengue virus serotypes have been reported, marked as DENV 1 – 4. One of those serotypes predominated the outbreaks in specific years [20]. Since the severity of an epidemic tends to be associated with its serotype, mathematical modelling can be a reliable tool to make hypotheses about the emergence of new strains.": this is incorrect. Disease severity is explained by the occurrence of secondary heterologous infections and inadequate medical interventions. Thus, the best way to predict severity is to conduct seroprevalence studies complemented by sentinel surveillance to identify different serotypes. I don't find any rationale in using this model to hypothesise new strains.

Authors’ Response to Reviewer 4 comment 3: We express our sincere appreciation for the thorough evaluation and valuable feedback provided by the esteemed reviewers.

The reviewer accurately highlights the crucial distinction between the severity of a disease and the severity of a disease outbreak, a differentiation we duly recognise. In referring to the severity of an epidemic in the context of serotype dominance, our intention is to encompass a comprehensive perspective, considering factors such as disease impact, transmissibility, economic and societal consequences, geographic reach, response capability, population susceptibility, and availability of vaccination and treatment resources, among other considerations (Thompson, R.N., Gilligan, C.A., and Cunniffe, N.J., 2020). It is the amalgamation of these factors that contributes to the overall assessment of the severity and repercussions of an epidemic.

While we hold the reviewer's perspective on the value of seroprevalence studies and sentinel surveillance in high regard, our approach centres on mathematical modelling as a complementary tool. This modelling perspective is employed to formulate hypotheses regarding the emergence of new strains in the broader context of disease outbreak severity. Our argument emphasises that mathematical modelling can be instrumental in hypothesising the emergence of new strains with different virulence. In other words, if disease occurrence exceeds the model's explanatory capacity, it may be attributable to the emergence of new strains with distinct virulence.

We acknowledge the significance of various methodological approaches in comprehending and managing diseases and appreciate the input provided by the reviewer. However, we contend that our emphasis on the utility of mathematical modelling, as elucidated in the provided reference, aligns well with the investigation of disease outbreak severity.

We remain open to further discussion and are prepared to address any additional concerns or queries raised by the reviewer.

---

## [Decision Letter · Decision Letter 3]

20 Feb 2024

Forecasting Weekly Dengue Incidence in Sri Lanka: Modified Autoregressive Integrated Moving Average Modeling Approach

PONE-D-23-16303R3

Dear Dr. Jayathilaka,

We’re pleased to inform you that your manuscript has been judged scientifically suitable for publication and will be formally accepted for publication once it meets all outstanding technical requirements.

Kind regards,

Mohamed R. Abonazel, Ph.D.

Academic Editor

PLOS ONE

Additional Editor Comments (optional):

Reviewers' comments:

Reviewer's Responses to Questions

**Comments to the Author**

1. If the authors have adequately addressed your comments raised in a previous round of review and you feel that this manuscript is now acceptable for publication, you may indicate that here to bypass the “Comments to the Author” section, enter your conflict of interest statement in the “Confidential to Editor” section, and submit your "Accept" recommendation.

Reviewer #3: All comments have been addressed

2. Is the manuscript technically sound, and do the data support the conclusions?

Reviewer #3: (No Response)

3. Has the statistical analysis been performed appropriately and rigorously? 

Reviewer #3: (No Response)

4. Have the authors made all data underlying the findings in their manuscript fully available?

Reviewer #3: (No Response)

5. Is the manuscript presented in an intelligible fashion and written in standard English?

Reviewer #3: (No Response)

6. Review Comments to the Author

Reviewer #3: (No Response)

7. PLOS authors have the option to publish the peer review history of their article (what does this mean?). If published, this will include your full peer review and any attached files.

Reviewer #3: **Yes: **Ross J. Gore
